# Efficient Continual Learning with Modular Networks and Task-Driven Priors

**Tom Veniat**
LIP6, Sorbonne Université, France
`tom.veniat@lip6.fr`

**Ludovic Denoyer & Marc'Aurelio Ranzato**
Facebook Artificial Intelligence Research
`{denoyer,ranzato}@fb.com`

## Abstract

Existing literature in Continual Learning (CL) has focused on overcoming catastrophic forgetting, the inability of the learner to recall how to perform tasks observed in the past. There are however other desirable properties of a CL system, such as the ability to transfer knowledge from previous tasks and to scale memory and compute sub-linearly with the number of tasks. Since most current benchmarks focus only on forgetting using short streams of tasks, we first propose a new suite of benchmarks to probe CL algorithms across these new axes. Finally, we introduce a new modular architecture, whose modules represent atomic skills that can be composed to perform a certain task. Learning a task reduces to figuring out which past modules to re-use, and which new modules to instantiate to solve the current task. Our learning algorithm leverages a task-driven prior over the exponential search space of all possible ways to combine modules, enabling efficient learning on long streams of tasks. Our experiments show that this modular architecture and learning algorithm perform competitively on widely used CL benchmarks while yielding superior performance on the more challenging benchmarks we introduce in this work. The Benchmark is publicly available at `https://github.com/facebookresearch/CTrLBenchmark`.

## 1 Introduction

Continual Learning (CL) is a learning framework whereby an agent learns through a sequence of tasks (Ring, 1994; Thrun, 1994; 1998), observing each task once and only once. Much of the focus of the CL literature has been on avoiding *catastrophic forgetting* (McClelland et al., 1995; McCloskey & Cohen, 1989; Goodfellow et al., 2013), the inability of the learner to recall how to perform a task learned in the past. In our view, remembering how to perform a previous task is particularly important because it promotes knowledge accrual and transfer. CL has then the potential to address one of the major limitations of modern machine learning: its reliance on large amounts of labeled data. An agent may learn well a new task even when provided with little labeled data if it can leverage the knowledge accrued while learning previous tasks.

Our first contribution is then to pinpoint general properties that a good CL learner should have, besides avoiding forgetting. In §3 we explain how the learner should be able to *transfer knowledge* from related tasks seen in the past. At the same time, the learner should be able to *scale sub-linearly* with the number of tasks, both in terms of memory and compute, when these are related.

Our second contribution is to introduce a new benchmark suite, dubbed CTrL, to test the above properties, since current benchmarks only focus on forgetting. For the sake of simplicity and as a first step towards a more holistic evaluation of CL models, in this work we restrict our attention to supervised learning tasks and basic transfer learning properties. Our experiments show that while commonly used benchmarks do not discriminate well between different approaches, our newly introduced benchmark let us dissect performance across several new dimensions of transfer and scalability (see Fig. 1 for instance), helping machine learning developers better understand the strengths and weaknesses of various approaches.

Our last contribution is a new model that is designed according to the above mentioned properties of CL methods. It is based on a modular neural network architecture (Eigen et al., 2014; Denoyer & Gallinari, 2015; Fernando et al., 2017; Li et al., 2019) with a novel task-driven prior (§4). Every

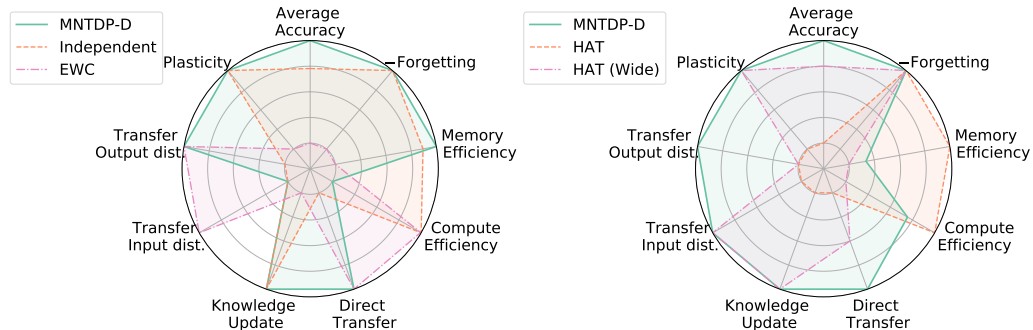

Figure 1: Comparison of various CL methods on the CTrL benchmark using Resnet (left) and Alexnet (right) backbones. MNTDP-D is our method. See Tab. 1 of §5.3 for details.

task is solved by the composition of a handful of neural modules which can be either borrowed from past tasks or freshly trained on the new task. In principle, modularization takes care of all the fundamental properties we care about, as i) by design there is no forgetting as modules from past tasks are not updated when new tasks arrive, ii) transfer is enabled via sharing modules across tasks, and iii) the overall model scales sublinearly with the number of tasks as long as similar tasks share modules. The key issue is how to efficiently select modules, as the search space grows exponentially in their number. In this work, we overcome this problem by leveraging a *data driven prior* over the space of possible architectures, which allows only local perturbations around the architecture of the previous task whose features best solve the current task (§4.2).

Our experiments in §5, which employ a stricter and more realistic evaluation protocol whereby streams are observed only once but data from each task can be played multiple times, show that this model performs at least as well as state-of-the-art methods on standard benchmarks, and much better on our new and more challenging benchmark, exhibiting better transfer and ability to scale to streams with a hundred tasks.

## 2 RELATED WORK

CL methods can be categorized into three main families of approaches. **Regularization** based methods use a single shared predictor across all tasks with the only exception that there can be a task-specific classification head depending on the setting. They rely on various regularization methods to prevent forgetting. Kirkpatrick et al. (2016); Schwarz et al. (2018) use an approximation of the Fisher Information matrix while Zenke et al. (2017) using the distance of each weight to its initialization as a measure of importance. These approaches work well in streams containing a limited number of tasks but will inevitably either forget or stop learning as streams grow in size and diversity (van de Ven & Tolias, 2019), due to their structural rigidity and fixed capacity.

Similarly, **rehearsal** based methods also share a single predictor across all tasks but attack forgetting by using rehearsal on samples from past tasks. For instance, Lopez-Paz & Ranzato (2017); Chaudhry et al. (2019b); Rolnick et al. (2019) store past samples in a replay buffer, while Shin et al. (2017) learn to generate new samples from the data distribution of previous tasks and Zhang et al. (2019) computes per-class prototypes. These methods share the same drawback of regularization methods: Their capacity is fixed and pre-determined which makes them ineffective at handling long streams.

Finally, approaches based on **evolving architectures** directly tackle the issue of the limited capacity by enabling the architecture to grow over time. Rusu et al. (2016) introduce a new network on each task, with connection to all previous layers, resulting in a network that grows super-linearly with the number of tasks. This issue was later addressed by Schwarz et al. (2018) who propose to distill the new network back to the original one after each task, henceforth yielding a fixed capacity predictor which is going to have severe limitations on long streams. Yoon et al. (2018); Hung et al. (2019) propose a heuristic algorithm to automatically add and prune weights.Mehta et al. (2020) propose a Bayesian approach adding an adaptive number of weights to each layer. Li et al. (2019) propose to softly select between reusing, adapting, and introducing a new module at every layer. Similarly, Xu & Zhu (2018) propose to add filters once a new task arrives using REINFORCE (Williams, 1992), leading to larger and larger networks even at inference time as time goes by. These two works are the most similar to ours, with the major difference that we restrict the search space over architectures,

enabling much better scaling to longer streams. While their search space (and RAM consumption) grows over time, ours is constant. Our approach is *modular*, and only a small (and constant) number of modules is employed for any given task both at training and test time. Non-modular approaches, like those relying on individual neuron gating (Adel et al., 2020; Serrà et al., 2018; Kessler et al., 2019), lack such runtime efficiency which limits their applicability to long streams. Rajasegaran et al. (2019); Fernando et al. (2017) propose to learn a modular architecture, where each task is identified by a path in a graph of modules like we do. However, they lack the prior over the search space. They both use random search which is rather inefficient as the number of modules grows.

There are of course other works that introduce new benchmarks for CL. Most recently, Wortsman et al. (2020) have proposed a stream with 2500 tasks all derived from MNIST permutations. Unfortunately, this may provide little insight in terms of how well models transfer knowledge across tasks. Other benchmarks like CORe50 (Lomonaco & Maltoni, 2017) and CUB-200 (Wah et al., 2011) are more realistic but do not enable precise assessment of how well models transfer and scale.

CL is also related to other learning paradigms, such as meta-learning (Finn et al., 2017; Nichol et al., 2018; Duan et al., 2016), but these only consider the problem of quickly adapting to a new task while in CL we are also interested in preventing forgetting and learning better over time. For instance, Alet et al. (2018) proposed a modular approach for robotic applications. However, only the performance on the last task was measured. There is also a body of literature on modular networks for multi-task and multi-domain learning (Ruder et al., 2019; Rebuffi et al., 2017; Zhao et al., 2020). The major differences are the static nature of the learning problem they consider and the lack of emphasis on scaling to a large number of tasks.

## 3 EVALUATING CONTINUAL LEARNING MODELS

Let us start with a general formalization of the CL framework. We assume that tasks arrive in sequence, and that each task is associated with an integer task descriptor $t = 1, 2, ...$ which corresponds to the order of the tasks. Task descriptors are provided to the learner both during training and test time. Each task is defined by a labeled dataset $\mathcal{D}^t$. We denote with $\mathcal{S}$ a sequence of such tasks. A predictor for a given task $t$ is denoted by $f : \mathcal{X}^t \times \mathbb{Z} \to \mathcal{Y}^t$. The predictor has internally some trainable parameters whose values depend on the stream of tasks $\mathcal{S}$ seen in the past, therefore the prediction is: $f(x, t|\mathcal{S})$. Notice that in general, different streams lead to different predictors for the same task: $f(x, t|\mathcal{S}) \neq f(x, t|\mathcal{S}')$.

**Desirable Properties of CL models And Metrics:**    Since we are focusing on supervised learning tasks, it is natural to evaluate models in terms of accuracy. We denote the prediction accuracy of the predictor $f$ as $\Delta(f(x, t|\mathcal{S}), y)$, where $x$ is the input, $t$ is the task descriptor of $x$, $\mathcal{S}$ is the stream of tasks seen by the learner and $y$ is the ground truth label.

In this work, we consider four major properties of a CL algorithm. First, the algorithm has to yield predictors that are accurate by the end of the learning experience. This is measured by their **average accuracy** at the end of the learning experience:

$$\mathcal{A}(\mathcal{S}) = \frac{1}{T} \sum_{t=1}^{T} \mathbb{E}_{(x,y)\sim\mathcal{D}^t}[\Delta(f(x, t|\mathcal{S} = 1, \dots, T), y)]. \tag{1}$$

Second, the CL algorithm should yield predictors that do not forget, i.e. that are able to perform a task seen in the past without significant loss of accuracy. **Forgetting** is defined as:

$$\mathcal{F}(\mathcal{S}) = \frac{1}{T-1} \sum_{t=1}^{T-1} \mathbb{E}_{(x,y)\sim\mathcal{D}^t}[\Delta(f(x, t|\mathcal{S} = 1, \dots, T), y) - \Delta(f(x, t|\mathcal{S}' = 1, \dots, t), y)] \tag{2}$$

This measure of forgetting has been called backward transfer (Lopez-Paz & Ranzato, 2017), and it measures the average loss of accuracy on a task at the end of training compared to when the task was just learned. Negative values indicate the model has been forgetting. Positive values indicate that the model has been improving on past tasks by learning subsequent tasks.

Third, the CL algorithm should yield predictors that are capable of transferring knowledge from past tasks when solving a new task. **Transfer** can be measured by:

$$\mathcal{T}(\mathcal{S}) = \mathbb{E}_{(x,y)\sim\mathcal{D}^T}[\Delta(f(x, T|\mathcal{S} = 1, \dots, T), y) - \Delta(f(x, T|\mathcal{S}' = T), y)] \tag{3}$$

which measures the difference of performance between a model that has learned through a whole sequence of tasks and a model that has learned the last task in isolation. We would expect this quantity to be positive if there exist previous tasks that are related to the last task. Negative values imply the model has suffered some form of interference or even lack of plasticity when the predictor has too little capacity left to learn the new task.

Finally, the CL algorithm has to yield predictors that **scale sublinearly** with the number of tasks both in terms of memory and compute. In order to quantify this, we simply report the total memory usage and compute by the end of the learning experience during training. We therefore include in the memory consumption everything a learner has to keep around to be able to continually learn (e.g., regularization parameters of EWC or the replay buffer for experience replay).

**Streams** The metrics introduced above can be applied to any stream of tasks. While current benchmarks are constructed to assess forgetting, they fall short at enabling a comprehensive evaluation of *transfer* and *scalability* because they do not control for task relatedness and they are composed of too few tasks. Therefore, we propose a new suite of streams. If $t$ is a task in the stream, we denote with $t^-$ and $t^+$ a task whose data is sampled from the same distribution as $t$, but with a much smaller or larger labeled dataset, respectively. Finally, $t'$ and $t''$ are tasks that are similar to task $t$, while we assume no relation between $t_i$ and $t_j$, for all $i \neq j$.

We consider five axes of transfer and define a stream for each of them. While other dimensions certainly exist, here we are focusing on basic properties that any desirable model should possess.

**Direct Transfer**: we define the stream $\mathcal{S}^- = (t_1^+, t_2, t_3, t_4, t_5, t_1^-)$ where the last task is the same as the first one but with much less data to learn from. This is useful to assess whether the learner can directly transfer knowledge from the relevant task.

**Knowledge Update**: we define the stream $\mathcal{S}^+ = (t_1^-, t_2, t_3, t_4, t_5, t_1^+)$ where the last task has much more data than the first task with intermediate tasks that are unrelated. In this case, there should not be much need to transfer anything from previous tasks, and the system can just use the last task to update its knowledge of the first task.

**Transfer to similar Input/Output Distributions**: we define two streams where the last task is similar to the first task but the input distribution changes $\mathcal{S}^{\text{in}} = (t_1, t_2, t_3, t_4, t_5, t_1')$, or the output distribution changes $\mathcal{S}^{\text{out}} = (t_1, t_2, t_3, t_4, t_5, t_1'')$.

**Plasticity:** this is a stream where all tasks are unrelated, $\mathcal{S}^{\text{pl}} = (t_1, t_2, t_3, t_4, t_5)$, which is useful to measure the "ability to still learn" and potential interference (erroneous transfer from unrelated tasks) when learning the last task.

All these tasks are evaluated using $\mathcal{T}$ in eq. 3. Other dimensions of transfer (e.g., transfer with compositional task descriptors or under noisy conditions) are avenues of future work. Finally, we evaluate scalability using $\mathcal{S}^{\text{long}}$, a stream with 100 tasks of various degrees of relatedness and with varying amounts of training data. See §5.1 for more details.

# 4 MODULAR NETWORKS WITH TASK DRIVEN PRIOR (MNTDP)

In this section we describe an approach, dubbed MNTDP, designed according to the properties discussed in §3. The class of predictor functions $f(x, t|\mathcal{S})$ we consider in this work is *modular*, in the sense that predictors are composed of modules that are potentially shared across different (but related) tasks. A module can be any parametric function, for instance, a ResNet block (He et al., 2015). The only restriction we use in this work is that all modules in a layer should differ only in terms of their actual parameter values, while modules across layers can implement different classes of functions. For instance, in the toy illustration of Fig. 2-A (see caption for details), there are two predictors, each composed of three modules (all ResNet blocks), the first one being shared.

## 4.1 TRAINING

Once the new task $t$ arrives, the algorithm follows three steps. First, it temporarily adds new randomly initialized modules at every layer (these are denoted by dashed boxes in Fig. 2-B) and it then defines a search space over all possible ways to combine modules. Second, it minimizes a loss function, which in our case is the cross-entropy loss, over both ways to combine modules and module parameters, see Fig. 2-C. Note that only the parameters of the newly added modules are subject to

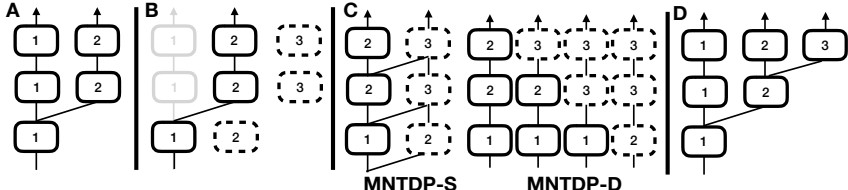

Figure 2: Toy illustration of the approach when each predictor is composed of only three modules and only two tasks have already been observed. **A)**: The predictor of the first task uses modules (1,1,1) (listing modules by increasing depth in the network) while the predictor of the second task uses modules (1,2,2); the first layer module is shared between the two predictors. **B)**: When a new task arrives, first we add one new randomly initialized module at each layer (the dashed modules). Second, we search for the most similar past task and retain only the corresponding architecture. In this case, the second task is most similar and therefore we remove (gray out) the modules used only by the predictor of the first task. **C)**: We train on the current task by learning both the best way to combine modules and their parameters. However, we restrict the search space. In this case, we only consider four possible compositions, all derived by perturbing the predictor of the second task. In the stochastic version (MNTDP-S), for every input a path (sequence of modules) is selected stochastically. Notice that the same module may contribute to multiple paths (e.g., the top-most layer with id 3). In the deterministic version instead (MNTDP-D), we train in parallel all paths and then select the best. Note that only the parameters of the newly added (dashed) modules are subject to learning. **D)**: Assuming that the best architecture found at the previous step is (1,2,3), module 3 at the top layer is added to the current library of modules.

training, de facto preventing forgetting of previous tasks by construction but also preventing positive backward transfer. Finally, it takes the resulting predictor for the current task and it adds the parameters of the newly added modules (if any) back to the existing library of module parameters, see Fig. 2-D.

Since predictors are uniquely identified by which modules compose them, they can also be described by the *path* in the grid of module parameters. We denote the $j$-th path in the graph by $\pi_j$. The parameters of the modules in path $\pi_j$ are denoted by $\theta(\pi_j)$. Note that in general $\theta(\pi_j) \cap \theta(\pi_i) \neq \emptyset$, for $i \neq j$ since some modules may be shared across two different paths.

Let $\Pi$ be the set of all possible paths in the graph. This has a size equal to the product of the number of modules at every layer, after adding the new randomly initialized modules. If $\Gamma$ is a distribution over $\Pi$ which is subject to learning (and initialized uniformly), then the loss function is:

$$\Gamma^*, \theta^* = \arg\min_{\theta, \Gamma} \mathbb{E}_{j \sim \Gamma, (x,y) \sim \mathcal{D}^t} \mathcal{L}(f(x, t | \mathcal{S}, \theta(\pi_j)), y) \tag{4}$$

where $f(x, t | \mathcal{S}, \theta(\pi_j)$ is the predictor using parameters $\theta(\pi_j)$, $\mathcal{L}$ is the loss, and the minimization over the parameters is limited to only the newly introduced modules. The resulting distribution $\Gamma^*$ is a delta distribution, assuming no ties between paths. Once the best path has been found and its parameters have been learned, the corresponding parameters of the new modules in the optimal path are added to the existing set of modules while the other ones are disregarded (Fig. 2-D). In this work, we consider two instances of the learning problem in eq. 4, which differ in a) how they optimize over paths and b) how they share parameters across modules.

**Stochastic version:** This algorithm alternates between one step of gradient descent over the paths via REINFORCE (Williams, 1992) as in (Veniat & Denoyer, 2018), and one step of gradient descent over the parameters for a given path. The distribution $\Gamma$ is modeled by a product of multinomial distributions, one for each layer of the model. These select one module at each layer, ultimately determining a particular path. Newly added modules may be shared across several paths which yields a model that can support several predictors while retaining a very parsimonious memory footprint thanks to parameter sharing. This version of the model, dubbed MNTDP-S, is outlined in the left part of Fig. 2-C and in Alg. 1 in the Appendix. In order to encourage the model to explore multiple paths, we use an entropy regularization on $\Gamma$ during training.

**Deterministic version:** This algorithm minimizes the objective over paths in eq. 4 via exhaustive search, see Alg. 2 in Appendix and the right part of Fig. 2-C. Here, paths do not share any newly added module and we train one independent network per path, and then select the path yielding the lowest loss on the validation set. While this requires much more memory, it may also lead to better

overall performance because each new module is cloned and trained just for a single path. Moreover, training predictors on each path can be trivially and cheaply parallelized on modern GPU devices.

## 4.2 DATA-DRIVEN PRIOR

Unfortunately, the algorithms as described above do not scale to a large number of tasks (and henceforth modules) because the search space grows exponentially. This is also the case for other evolving architecture approaches proposed in the literature (Li et al., 2019; Rajasegaran et al., 2019; Yoon et al., 2018; Xu & Zhu, 2018) as discussed in §2.

If there were $N$ modules per layer and $L$ layers, the search space would have size $N^L$. In order to restrict the search space, we only allow paths that branch to the right: A newly added module at layer $l$ can only connect to another newly added module at layer $l + 1$, but it cannot connect to an already trained module at layer $l + 1$. The underlying assumption is that for most tasks we expect changes in the output distribution as opposed to the input distribution, and therefore if tasks are related, the base trunk is a good candidate for being shared. We will see in §5.3 what happens when this assumption is not satisfied, e.g., when applying this to $\mathcal{S}^{\mathrm{in}}$.

To further restrict the search space we employ a *data-driven prior*. The intuition is to limit the search space to perturbations of the path corresponding to the past task (or to the top-$k$ paths) that is the most similar to the current task. There are several methods to assess which task is the closest to the current task without accessing data from past tasks and also different ways to perturb a path. We propose a very simple approach, but others could have been used. We take the predictors from all the past tasks and select the path that yields the best nearest neighbor classification accuracy when feeding data from the current task using the features just before the classification head. This process is shown in Fig. 2-B. The search space is reduced from $T^L$ to only $L$, and $\Gamma$ of eq. 4 is allowed to have non-zero support only in this restricted search space, yielding a much lower computational and memory footprint which is *constant* with respect to the number of tasks. The designer of the model has now direct control (by varying $k$, for instance) over the trade-off between accuracy and computational/memory budget.

By construction, the model does not forget, because we do not update modules of previous tasks. The model can transfer well because it can re-use modules from related tasks encountered in the past while not being constrained in terms of its capacity. And finally, the model scales sub-linearly in the number of tasks because modules can be shared across similar tasks. We will validate empirically in §5.3 whether the choice of the restricted search space works well in practice.

## 5 EXPERIMENTS

In this section we first introduce our benchmark in §5.1 and the modeling details §5.2, and then report results both on standard benchmarks as well as ours in §5.3.[1]

## 5.1 THE CTRL BENCHMARK

The CTrL (Continual Transfer Learning) benchmark is a collection of streams of tasks built over seven popular computer vision datasets, namely: CIFAR10 and CIFAR100 (Krizhevsky, 2009), DTD (Cimpoi et al., 2014), SVHN (Netzer et al., 2011), MNIST (LeCun et al., 1998), Rainbow-MNIST (Finn et al., 2019) and Fashion MNIST (Xiao et al., 2017); see Table 3 in Appendix for basic statistics. These datasets are desirable because they are diverse (hence tasks derived from some of these datasets can be considered unrelated), they have a fairly large number of training examples to simulate tasks that do not need to transfer, and they have low spatial resolution enabling fast evaluation. CTrL is designed according to the methodology described in §3, to enable evaluation of various transfer learning properties and the ability of models to scale to a large number of tasks. Each task consists of a training, validation, and test datasets corresponding to a 5-way and 10-way classification problem for the transfer streams and the long stream, respectively. The last task of $\mathcal{S}^{\mathrm{out}}$ consists of a shuffling of the output labels of the first task. The last task of $\mathcal{S}^{\mathrm{in}}$ is the same as its first task except that MNIST images have a different background color. $\mathcal{S}^{\mathrm{long}}$ has 100 tasks, and it is

---

[1]CTrL source code available at `https://github.com/facebookresearch/CTrLBenchmark`. Pytorch implementation of the experiments available here: `https://github.com/TomVeniat/MNTDP`.

constructed by first sampling a dataset, then 5 classes at random, and finally the amount of training data from a distribution that favors small tasks by the end of the learning experience. See Tab. 5 and 6 in Appendix for details. Therefore, $\mathcal{S}^{\text{long}}$ tests not only the ability of a model to scale to a relatively large number of tasks but also to transfer more efficiently with age.

All images have been rescaled to 32x32 pixels in RGB color format, and per-channel normalized using statistics computed on the training set of each task. During training, we perform data augmentation by using random crops (4 pixels padding and 32x32 crops) and random horizontal reflection. Please, refer to Appendix A for further details.

## 5.2 METHODOLOGY AND MODELING DETAILS

Models learn over each task in sequence; data from each task can be replayed several times but *each stream is observed only once*. Since each task has a validation dataset, hyper-parameters (e.g., learning rate and number of weight updates) are task-specific and they are cross-validated on the validation set of each task. Once the learning experience ends, we test the resulting predictor on the test sets of all the tasks. Notice that this is a stricter paradigm than what is usually employed in the literature (Kirkpatrick et al., 2016), where hyper-parameters are set at the stream level (by replaying the stream several times). Our model selection criterion is more realistic because it does not assume that the learner has access to future tasks when cross-validating on the current task, and this is more consistent with the CL's assumptions of operating on a stream of data.

All models use the same backbone. Unless otherwise specified, this is a small variant of the ResNet-18 architecture which is divided into 7 modules; please, refer to Table 8 for details of how many layers and parameters each block contains. Each predictor is trained by minimizing the cross-entropy loss with a small L2 weight decay on the parameters. In our experiments, MNTDP adds 7 new randomly initialized modules, one for every block. The search space does not allow connecting old blocks from new blocks, and it considers two scenarios: using old blocks from the past task that is deemed most similar ($k = 1$, the default setting) or considering the whole set of old blocks ($k =$ all) resulting in a much larger search space.

We compare to several baselines: **Independent Models** which instantiates a randomly initialized predictor for every task (as many paths as tasks without any module overlap), 2) **Finetuning** which trains a single model to solve all the tasks without any regularization and initializes from the model of the previous task (a single path shared across all tasks), 3) **New-Head** which also shares the trunk of the network across all tasks but not the classification head which is task-specific, 4) **New-Leg** which shares all layers across tasks except for the very first input layer which is task-specific, 5) **EWC** (Kirkpatrick et al., 2016) which is like "finetuning" but with a regularizer to alleviate forgetting, 6) **Experience Replay** (Chaudhry et al., 2019b) which is like finetuning except that the model has access to some samples from the past tasks to rehearse and alleviate forgetting (we use 15 samples per class to obtain a memory consumption similar to other baselines), 7) **Progressive Neural Networks** (PNNs) (Rusu et al., 2016) which adds both a new module at every layer as well as lateral connections once a new task arrives. 8) **HAT** (Serrà et al., 2018): which learns an attention mask over the parameters of the backbone network for each task. Since HAT's open-source implementation uses AlexNet (Krizhevsky et al., 2012) as a backbone, we also implemented a version of MNTDP using AlexNet for a fair comparison. Moreover, we considered two versions of HAT, the default as provided by the authors and a version, dubbed HAT-wide, that is as wide as our final MNTDP model (or as wide as we can fit into GPU memory). 9) **DEN** (Yoon et al., 2018) and 10) **RCL** (Xu & Zhu, 2018) which both evolve architectures. For these two models since there is no publicly available implementation with CNNs, we only report their performance on Permuted-MNIIST using fully connected nets (see Fig. 3a and Tab. 11 in Appendix E.2).

We report performance across different axes as discussed in §3: Average accuracy as in eq. 1, forgetting as in eq. 2, transfer as in eq. 3 and applicable only to the transfer streams, "Mem." [MB] which refers to the average memory consumed by the end of training, and "Flops" [T] which corresponds to the average amount of computation used by the end of training.

## 5.3 RESULTS

**Existing benchmarks:** In Fig. 3 we compare MNTDP against several baselines on two standard streams with 10 tasks, Permuted MNIST, and Split CIFAR100. We observe that all models do fairly

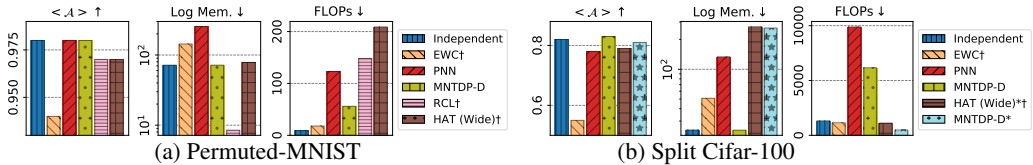

(a) Permuted-MNIST  (b) Split Cifar-100

Figure 3: Results on standard continual learning streams. * denotes an Alexnet Backbone. † correspond to models cross-validated at the stream-level, a setting that favors them over the other methods which are cross-validated at the task-level. Detailed results are presented in Appendix E.2.

|  | $< \mathcal{A} >$ | $< \mathcal{F} >$ | Mem. | FLOPs | $\mathcal{T}(\mathcal{S}^-)$ | $\mathcal{T}(\mathcal{S}^+)$ | $\mathcal{T}(\mathcal{S}^{in})$ | $\mathcal{T}(\mathcal{S}^{out})$ | $\mathcal{T}(\mathcal{S}^{pl})$ |
|---|---|---|---|---|---|---|---|---|---|
| Independent | 0.58 | **0.0** | 14.1 | 308 | 0.0 | **0.0** | 0.0 | 0.0 | **0.0** |
| Finetune | 0.19 | -0.3 | **2.4** | **284** | 0.0 | -0.1 | -0.0 | -0.0 | -0.1 |
| New-head | 0.48 | **0.0** | 2.5 | 307 | **0.4** | -0.3 | -0.2 | **0.3** | -0.4 |
| New-leg | 0.41 | **0.0** | 2.5 | 366 | 0.3 | -0.3 | **0.4** | -0.1 | -0.4 |
| Online EWC † | 0.43 | -0.1 | 7.3 | 310 | 0.3 | -0.3 | 0.3 | **0.3** | -0.4 |
| ER | 0.44 | -0.1 | 13.1 | 604 | 0.0 | -0.2 | 0.0 | 0.1 | -0.2 |
| PNN | 0.57 | **0.0** | 48.2 | 1459 | 0.3 | -0.2 | 0.1 | 0.2 | -0.1 |
| MNTDP-S | 0.59 | **0.0** | 11.7 | 363 | **0.4** | -0.1 | 0.0 | **0.3** | -0.1 |
| MNTDP-D | **0.64** | **0.0** | 11.6 | 1512 | **0.4** | **0.0** | 0.0 | **0.3** | **-0.0** |
| MNTDP-D* | 0.62 | 0.0 | 140.7 | 115 | 0.3 | -0.1 | 0.1 | 0.3 | -0.1 |
| HAT*† | 0.58 | -0.0 | 26.6 | 45 | 0.1 | -0.2 | 0.0 | 0.1 | -0.2 |
| HAT (Wide)*† | 0.61 | 0.0 | 163.9 | 274 | 0.2 | -0.1 | 0.1 | 0.1 | -0.1 |

Table 1: Aggregated results on the transfer streams over multiple relevant baselines (complete table with more baselines provided in Appendix E). * correspond to models using an Alexnet backbone.

well, with EWC falling a bit behind the others in terms of average accuracy. PNNs has good average accuracy but requires more memory and compute. Compared to MNTDP, both RCL and HAT have lower average accuracy and require more compute. MNTDP-D yields the best average accuracy, but requires more computation than "independent models"; notice however that its wall-clock training time is actually the same as "independent models" since all candidate paths (seven in our case) can be trained in parallel on modern GPU devices. In fact, it turns out that on these standard streams MNTDP trivially reduces to "independent models" without any module sharing, since each task is fairly distinct and has a relatively large amount of data. It is therefore not possible to assess how well models can transfer knowledge across tasks, nor it is possible to assess how well models scale. Fortunately, we can leverage the CTrL benchmark to better study these properties, as described next.

**CTrL:** We first evaluate models in terms of their ability to transfer by evaluating them on the streams $\mathcal{S}^-, \mathcal{S}^+, \mathcal{S}^{in}, \mathcal{S}^{out}$ and $\mathcal{S}^{pl}$ introduced in §3. Tab. 1 shows that "independent models" is again a strong baseline, because even on the first four streams, all tasks except the last one are unrelated and therefore instantiating an independent model is optimal. However, MNTDP yields the best average accuracy overall. MNTDP-D achieves the best transfer on streams $\mathcal{S}^-, \mathcal{S}^+, \mathcal{S}^{out}$ and $\mathcal{S}^{pl}$, and indeed it discovers the correct path in each of these cases (e.g., it discovers to reuse the path of the first task when learning on $\mathcal{S}^-$ and to just swap the top modules when learning the last task on $\mathcal{S}^+$). Examples of discovered path are presented in appendix E.4. MNTDP underperforms on $\mathcal{S}^{in}$ because its prior does not match the data distribution, since in this case, it is the input distribution that has changed but swapping the first module is out of MNTDP search space. This highlights the importance of the choice of prior for this algorithm. In general, MNTDP offers a clear trade-off between accuracy, i.e. how broad the prior is which determines how many paths can be evaluated, and memory/compute budget. Computationally MNTDP-D is the most demanding, but in practice its wall clock time is comparable to "independent" because GPU devices can store in memory all the paths (in our case, seven) and efficiently train them all in parallel. We observe also that MNTDP-S has a clear advantage in terms of compute at the expense of a lower overall average accuracy, as sharing modules across paths during training can lead to sub-optimal convergence.

Overall, MNTDP has a much higher average accuracy than methods with a fixed capacity. It also beats PNNs, which seems to struggle with interference when learning on $\mathcal{S}^{pl}$, as all new modules connect to all old modules which are irrelevant for the last task. Moreover, PNNs uses much more

| | $< \mathcal{A} >$ | $< \mathcal{F} >$ | Mem. | PFLOPs |
|---|---|---|---|---|
| Independent | 0.57 | 0.0 | 243 | 4 |
| Finetune | 0.20 | -0.4 | 2 | 5 |
| New-head | 0.43 | 0.0 | 3 | 6 |
| On. EWC† | 0.27 | -0.3 | 7 | 4 |
| MNTDP-S | 0.68 | 0.0 | 159 | 5 |
| MNTDP-D | 0.75 | 0.0 | 102 | 26 |
| MNTDP-D* | 0.75 | 0.0 | 1782 | 3 |
| HAT*† | 0.24 | -0.1 | 32 | ≈0 |
| HAT*† (Wide) | 0.32 | 0.0 | 285 | 1 |

Figure 4: Evolution of $< \mathcal{A} >$ and Mem. on $\mathcal{S}^{\text{long}}$.

Table 2: Results on the long evaluation stream. * correspond to models using an Alexnet backbone. See Tab. 18 for more baselines and error bars.

memory. "New-leg" and "new-head" models perform very well only when the tasks in the stream match their assumptions, showing the advantage of the adaptive prior of MNTDP. Finally, EWC shows great transfer on $\mathcal{S}^{\text{-}}, \mathcal{S}^{\text{in}}, \mathcal{S}^{\text{out}}$, which probe the ability of the model to retain information. However, it fails at $\mathcal{S}^{+}, \mathcal{S}^{\text{pl}}$ that require additional capacity allocation to learn a new task. Fig. 1 gives a holistic view by reporting the normalized performance across all these dimensions.

We conclude by reporting results on $\mathcal{S}^{\text{long}}$ composed of 100 tasks. Tab. 2 reports the results of all the approaches we could train without running into out-of-memory. MNTDP-D yields an absolute 18% improvement over the baseline *independent model* while using less than half of its memory, thanks to the discovery of many paths with shared modules. Its actual runtime is close to *independent model* because of GPU parallel computing. To match the capacity of MNTDP, we scale HAT's backbone to the maximal size that can fit in a Titan X GPU Memory (6.5x, wide version). The wider architecture greatly increases inference time in later tasks (see also discussion on memory complexity at test time in Appendix D), while our modular approach uses the same backbone for every task and yet it achieves better performance. Fig. 4 shows the average accuracy up to the current task over time. MNTDP-D attains the best performance while growing sublinearly in terms of memory usage. Methods that do not evolve their architecture, like EWC, greatly suffer in terms of average accuracy.

**Ablation:** We first study the importance of the prior. Instead of selecting the nearest neighbor path, we pick one path corresponding to one of the previous tasks at random. In this case, $\mathcal{T}(\mathcal{S}^{\text{-}})$ decreases from 0 to $-0.2$ and $\mathcal{T}(\mathcal{S}^{out})$ goes from 0 to $-0.3$. With a random path, MNTDP learns not to share any module, demonstrating that it is indeed important to form a good prior over the search space. Appendix E reports additional results demonstrating how on small streams MNTDP is robust to the choice of $k$ in the prior since we attain similar performance using $k = 1$ and $k = all$, although only $k = 1$ let us scale to $\mathcal{S}^{\text{long}}$. Finally, we explore the robustness to the number of modules by splitting each module in two, yielding a total of 10 modules per path, and by merging adjacent modules yielding a total of 3 modules for the same overall number of parameters. We find that $\mathcal{T}(\mathcal{S}^{\text{out}})$ decreases from 0 to -0.1, with a 9% decrease on $t_1^-$, when the number of modules decreases but stays the same when the number of modules increases, suggesting that the algorithm has to have a sufficient number of modules to flexibly grow.

## 6 CONCLUSIONS

We introduced a new benchmark to enable a more comprehensive evaluation of CL algorithms, not only in terms of average accuracy and forgetting but also knowledge transfer and scaling. We have also proposed a modular network that can gracefully scale thanks to an adaptive prior over the search space of possible ways to connect modules. Our experiments show that our approach yields a very desirable trade-off between accuracy and compute/memory usage, suggesting that modularization in a restricted search space is a promising avenue of investigation for continual learning and knowledge transfer.

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

# A DATASETS AND STREAMS

The CTrL benchmark is built using the standard datasets listed in Tab. 3. Some examples from these datasets are shown in Fig. 5. The composition of the different tasks is given in Tab. 4 and an instance of the long stream is presented in Tab. 5 and 6.

The tasks in $\mathcal{S}^-$, $\mathcal{S}^+$, $\mathcal{S}^{in}$, $\mathcal{S}^{out}$ and $\mathcal{S}^{pl}$ are all 10-way classification tasks. In $\mathcal{S}^-$, the first task has 4000 training examples while the last one which is the same as the first task, has only 400. The vice versa is true for $\mathcal{S}^+$ instead. The last task of $\mathcal{S}^{in}$ is the same as the first task, except that the background color of the MNIST digit is different. The last task of $\mathcal{S}^{out}$ is the same as the first task, except that label ids have been shuffled, therefore, if "horse" was associated to label id 3 in the first task, it is now associated to label id 5 in the last task.

The $\mathcal{S}^{long}$ stream is composed of both *large* and *small* tasks that have 5000 (or whatever is the maximum available) and 25 training examples, respectively. Each task is built by choosing one of the datasets at random, and 5 categories at random in this dataset. During task 1-33, the fraction of small tasks is 50%, this increases to 75% for tasks 34-66, and to 100% for tasks 67-100. This is a challenging setting allowing to assess not only scalability, but also transfer ability and sample efficiency of a learner.

Scripts to rebuild the given streams and evaluate models will be released upon publication.

| Dataset | no. classes | training | validation | testing |
|---|---|---|---|---|
| CIFAR-100 | 100 | 40000 | 10000 | 10000 |
| CIFAR-10 | 10 | 40000 | 10000 | 10000 |
| D. Textures | 47 | 1880 | 1880 | 1880 |
| SVHN | 10 | 47217 | 26040 | 26032 |
| MNIST | 10 | 50000 | 10000 | 10000 |
| Fashion-MNIST | 10 | 50000 | 10000 | 10000 |
| Rainbow-MNIST | 10 | 50000 | 10000 | 10000 |

Table 3: Datasets used in the CTrL benchmark.

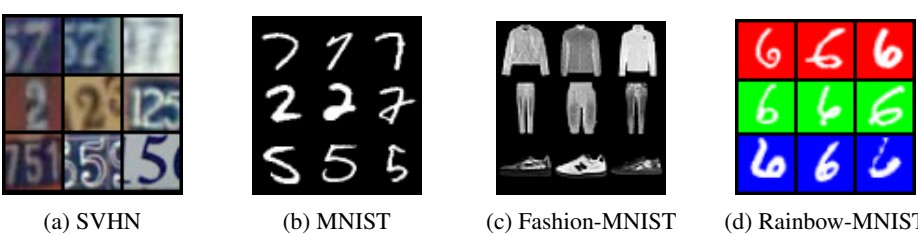

|   (a) SVHN   |   (b) MNIST   |   (c) Fashion-MNIST   |   (d) Rainbow-MNIST   |

Figure 5: Some datasets used in the CTrL benchmark.

| Stream | | $T_1$ | $T_2$ | $T_3$ | $T_4$ | $T_5$ | $T_6$ |
|---|---|---|---|---|---|---|---|
| | **Datasets** | Cifar-10 | MNIST | DTD | F-MNIST | SVHN | Cifar-10 |
| $\mathcal{S}^-$ | **# Train Samples** | 4000 | 400 | 400 | 400 | 400 | 400 |
| | **# Val Samples** | 2000 | 200 | 200 | 200 | 200 | 200 |
| | **Datasets** | Cifar-10 | MNIST | DTD | F-MNIST | SVHN | Cifar-10 |
| $\mathcal{S}^+$ | **# Train Samples** | 400 | 400 | 400 | 400 | 400 | 4000 |
| | **# Val Samples** | 200 | 200 | 200 | 200 | 200 | 2000 |
| | **Datasets** | R-MNIST | Cifar-10 | DTD | F-MNIST | SVHN | R-MNIST |
| $\mathcal{S}^{\text{in}}$ | **# Train Samples** | 4000 | 400 | 400 | 400 | 400 | 50 |
| | **# Val Samples** | 2000 | 200 | 200 | 200 | 200 | 30 |
| | **Datasets** | Cifar-10 | MNIST | DTD | F-MNIST | SVHN | Cifar-10 |
| $\mathcal{S}^{\text{out}}$ | **# Train Samples** | 4000 | 400 | 400 | 400 | 400 | 400 |
| | **# Val Samples** | 2000 | 200 | 200 | 200 | 200 | 200 |
| | **Datasets** | MNIST | DTD | F-MNIST | SVHN | Cifar-10 | |
| $\mathcal{S}^{\text{pl}}$ | **# Train Samples** | 400 | 400 | 400 | 400 | 4000 | |
| | **# Val Samples** | 200 | 200 | 200 | 200 | 2000 | |

Table 4: Details of the streams used to evaluate the transfer properties of the learner. F-MNIST is Fashion-MNIST and R-MNIST is a variant of Rainbow-MNIST, using only different background colors and keepingthe original scale and rotation of the digits

| Task id | Dataset | Classes | # Train | # Val | # Test |
|---|---|---|---|---|---|
| 1 | mnist | [6, 3, 7, 5, 0] | 25 | 15 | 4830 |
| 2 | svhn | [2, 1, 9, 0, 7] | 25 | 15 | 5000 |
| 3 | svhn | [2, 0, 6, 1, 5] | 5000 | 2500 | 5000 |
| 4 | svhn | [1, 5, 0, 7, 4] | 25 | 15 | 5000 |
| 5 | fashion-mnist | [T-shirt/top, Pullover, Trouser, Sandal, Sneaker] | 25 | 15 | 5000 |
| 6 | fashion-mnist | [Shirt, Ankle boot, Sandal, Pullover, T-shirt/... | 5000 | 2500 | 5000 |
| 7 | svhn | [3, 1, 7, 6, 9] | 25 | 15 | 5000 |
| 8 | cifar100 | [spider, maple\_tree, tulip, leopard, lizard] | 25 | 15 | 500 |
| 9 | cifar10 | [frog, automobile, airplane, cat, horse] | 25 | 15 | 5000 |
| 10 | fashion-mnist | [Ankle boot, Bag, T-shirt/top, Shirt, Pullover] | 25 | 15 | 5000 |
| 11 | mnist | [4, 8, 7, 6, 3] | 5000 | 2500 | 4914 |
| 12 | cifar10 | [automobile, truck, dog, horse, deer] | 5000 | 2500 | 5000 |
| 13 | cifar100 | [sea, forest, bear, chimpanzee, dinosaur] | 25 | 15 | 500 |
| 14 | mnist | [3, 2, 9, 1, 7] | 25 | 15 | 5000 |
| 15 | fashion-mnist | [Bag, Ankle boot, Trouser, Shirt, Dress] | 25 | 15 | 5000 |
| 16 | cifar10 | [frog, cat, horse, airplane, deer] | 25 | 15 | 5000 |
| 17 | cifar10 | [bird, frog, ship, truck, automobile] | 5000 | 2500 | 5000 |
| 18 | svhn | [0, 4, 7, 5, 6] | 5000 | 2500 | 5000 |
| 19 | mnist | [6, 5, 9, 4, 8] | 5000 | 2500 | 4806 |
| 20 | mnist | [8, 5, 6, 4, 9] | 5000 | 2500 | 4806 |
| 21 | cifar100 | [sea, pear, house, spider, aquarium\_fish] | 25 | 15 | 500 |
| 22 | cifar100 | [kangaroo, ray, tank, crocodile, table] | 2250 | 250 | 500 |
| 23 | cifar100 | [trout, rose, pear, lizard, baby] | 25 | 15 | 500 |
| 24 | svhn | [3, 2, 8, 1, 5] | 5000 | 2500 | 5000 |
| 25 | cifar100 | [skyscraper, bear, rocket, tank, spider] | 25 | 15 | 500 |
| 26 | cifar100 | [telephone, porcupine, flatfish, plate, shrew] | 2250 | 250 | 500 |
| 27 | cifar100 | [lawn\_mower, crocodile, tiger, bed, bear] | 25 | 15 | 500 |
| 28 | svhn | [3, 7, 1, 5, 6] | 25 | 15 | 5000 |
| 29 | fashion-mnist | [Ankle boot, Sneaker, T-shirt/top, Coat, Bag] | 5000 | 2500 | 5000 |
| 30 | mnist | [6, 9, 0, 3, 7] | 5000 | 2500 | 4938 |
| 31 | cifar10 | [automobile, truck, deer, bird, dog] | 25 | 15 | 5000 |
| 32 | cifar10 | [dog, airplane, frog, deer, automobile] | 5000 | 2500 | 5000 |
| 33 | svhn | [1, 9, 5, 3, 6] | 5000 | 2500 | 5000 |
| 34 | cifar100 | [whale, orange, chimpanzee, poppy, sweet\_pepper] | 25 | 15 | 500 |
| 35 | cifar100 | [worm, camel, bus, keyboard, spider] | 25 | 15 | 500 |
| 36 | fashion-mnist | [T-shirt/top, Coat, Ankle boot, Shirt, Dress] | 25 | 15 | 5000 |
| 37 | cifar10 | [dog, deer, ship, truck, cat] | 25 | 15 | 5000 |
| 38 | cifar10 | [cat, dog, airplane, ship, deer] | 5000 | 2500 | 5000 |
| 39 | svhn | [7, 6, 4, 2, 9] | 25 | 15 | 5000 |
| 40 | mnist | [9, 7, 1, 3, 2] | 25 | 15 | 5000 |
| 41 | cifar100 | [mushroom, butterfly, bed, boy, motorcycle] | 25 | 15 | 500 |
| 42 | fashion-mnist | [Shirt, Pullover, Bag, Sandal, T-shirt/top] | 25 | 15 | 5000 |
| 43 | cifar100 | [rabbit, bear, aquarium\_fish, bee, bowl] | 25 | 15 | 500 |
| 44 | fashion-mnist | [Coat, T-shirt/top, Pullover, Shirt, Sandal] | 25 | 15 | 5000 |
| 45 | fashion-mnist | [Pullover, Dress, Coat, Shirt, Sandal] | 25 | 15 | 5000 |
| 46 | mnist | [3, 9, 7, 6, 4] | 25 | 15 | 4940 |
| 47 | cifar10 | [deer, bird, dog, automobile, frog] | 25 | 15 | 5000 |
| 48 | svhn | [8, 7, 1, 0, 4] | 25 | 15 | 5000 |
| 49 | cifar100 | [forest, skunk, poppy, bridge, sweet\_pepper] | 2250 | 250 | 500 |
| 50 | cifar100 | [caterpillar, can, motorcycle, rabbit, wardrobe] | 25 | 15 | 500 |

Table 5: Details of the tasks used in $\mathcal{S}^{\text{long}}$, part 1.

| Task id | Dataset | Classes | # Train | # Val | # Test |
|---|---|---|---|---|---|
| 51 | cifar100 | [trout, mountain, kangaroo, pine\_tree, bee] | 25 | 15 | 500 |
| 52 | cifar100 | [clock, fox, castle, bus, willow\_tree] | 25 | 15 | 500 |
| 53 | cifar10 | [cat, airplane, dog, ship, truck] | 25 | 15 | 5000 |
| 54 | mnist | [9, 7, 8, 1, 5] | 25 | 15 | 4866 |
| 55 | fashion-mnist | [Bag, T-shirt/top, Sandal, Shirt, Dress] | 25 | 15 | 5000 |
| 56 | fashion-mnist | [Sneaker, Ankle boot, Coat, Sandal, Trouser] | 5000 | 2500 | 5000 |
| 57 | mnist | [1, 4, 3, 9, 7] | 25 | 15 | 4982 |
| 58 | cifar10 | [truck, automobile, frog, ship, dog] | 25 | 15 | 5000 |
| 59 | mnist | [7, 2, 8, 5, 4] | 25 | 15 | 4848 |
| 60 | mnist | [2, 8, 9, 1, 7] | 5000 | 2500 | 4974 |
| 61 | svhn | [9, 5, 1, 8, 6] | 25 | 15 | 5000 |
| 62 | mnist | [1, 8, 7, 4, 5] | 25 | 15 | 4848 |
| 63 | cifar10 | [truck, dog, bird, automobile, airplane] | 25 | 15 | 5000 |
| 64 | mnist | [8, 4, 3, 7, 6] | 25 | 15 | 4914 |
| 65 | svhn | [3, 5, 7, 2, 1] | 25 | 15 | 5000 |
| 66 | cifar100 | [otter, camel, bee, road, poppy] | 25 | 15 | 500 |
| 67 | svhn | [4, 2, 1, 8, 7] | 25 | 15 | 5000 |
| 68 | mnist | [3, 7, 6, 8, 9] | 25 | 15 | 4932 |
| 69 | fashion-mnist | [Pullover, Sneaker, Trouser, Dress, Sandal] | 25 | 15 | 5000 |
| 70 | svhn | [5, 0, 7, 2, 3] | 25 | 15 | 5000 |
| 71 | svhn | [9, 6, 2, 4, 8] | 25 | 15 | 5000 |
| 72 | mnist | [7, 1, 2, 0, 6] | 25 | 15 | 4938 |
| 73 | cifar10 | [dog, automobile, ship, airplane, cat] | 25 | 15 | 5000 |
| 74 | mnist | [0, 7, 6, 2, 4] | 25 | 15 | 4920 |
| 75 | cifar10 | [bird, deer, airplane, dog, ship] | 25 | 15 | 5000 |
| 76 | cifar100 | [mountain, bicycle, caterpillar, spider, possum] | 25 | 15 | 500 |
| 77 | svhn | [8, 3, 4, 0, 6] | 25 | 15 | 5000 |
| 78 | svhn | [1, 5, 9, 0, 8] | 25 | 15 | 5000 |
| 79 | cifar100 | [can, dolphin, house, pickup\_truck, crab] | 25 | 15 | 500 |
| 80 | cifar100 | [squirrel, possum, crocodile, mountain, hamster] | 25 | 15 | 500 |
| 81 | mnist | [7, 0, 1, 6, 2] | 25 | 15 | 4938 |
| 82 | fashion-mnist | [T-shirt/top, Dress, Trouser, Shirt, Sneaker] | 25 | 15 | 5000 |
| 83 | cifar10 | [cat, frog, automobile, dog, airplane] | 25 | 15 | 5000 |
| 84 | cifar10 | [automobile, cat, dog, ship, horse] | 25 | 15 | 5000 |
| 85 | cifar100 | [cup, otter, orchid, kangaroo, rose] | 25 | 15 | 500 |
| 86 | mnist | [1, 5, 7, 2, 9] | 25 | 15 | 4892 |
| 87 | svhn | [6, 5, 3, 2, 7] | 25 | 15 | 5000 |
| 88 | cifar10 | [dog, deer, cat, frog, bird] | 25 | 15 | 5000 |
| 89 | mnist | [6, 2, 5, 9, 4] | 25 | 15 | 4832 |
| 90 | cifar100 | [pear, rocket, sea, road, orange] | 25 | 15 | 500 |
| 91 | svhn | [0, 8, 4, 6, 1] | 25 | 15 | 5000 |
| 92 | cifar10 | [truck, horse, ship, deer, dog] | 25 | 15 | 5000 |
| 93 | mnist | [5, 8, 6, 4, 3] | 25 | 15 | 4806 |
| 94 | svhn | [2, 6, 3, 4, 1] | 25 | 15 | 5000 |
| 95 | fashion-mnist | [Bag, Trouser, Sneaker, Ankle boot, Sandal] | 25 | 15 | 5000 |
| 96 | svhn | [7, 9, 1, 5, 8] | 25 | 15 | 5000 |
| 97 | cifar100 | [lamp, otter, skyscraper, sea, raccoon] | 25 | 15 | 500 |
| 98 | cifar100 | [clock, flatfish, snake, can, man] | 25 | 15 | 500 |
| 99 | svhn | [6, 3, 0, 8, 7] | 25 | 15 | 5000 |
| 100 | fashion-mnist | [Shirt, Coat, Dress, Sandal, Pullover] | 25 | 15 | 5000 |

Table 6: Details of the task in $\mathcal{S}^{\text{long}}$, part 2.

## B  LEARNING ALGORITHM

We provide a more complete description of the two learning algorithms for MNTDP-S and MNTDP-D. In the deterministic case, all the architectures (given how we restrict the search space, we have 7 of them) are trained over the training set, and the best path is retained based on its score on the validation set. To avoid overfitting when using MNDTDP-S, we split the training set into two halves $\mathcal{D}_1^t$ and $\mathcal{D}_2^t$. The first part is used to update the module parameters $\theta$, while the second is used to update the parameters in the distribution over paths, $\Gamma$. If both sets of parameters were trained on the same dataset, $\Gamma$ would favor paths prone to overfitting since they will results in a large decrease of its training loss and therefore a larger reward. When they are trained on different sets, $\Gamma$ has to select paths with a reasonable amount of free parameters, allowing $\theta$ to learn and generalize enough to decrease the loss on both training sets. Then, the most promising architecture is selected based on $\arg\max \Gamma$, and fine-tuned over the whole $\mathcal{D}^t$. In this case, the validation set is used only for hyper-parameters selection.

---

**Algorithm 1:** MNTDP-S algorithm.

1 **Data:** Dataset of task $t$: $\mathcal{D}^t$.;
2 **Past predictors:** $f(x, j|\mathcal{S})$ for $j = 1, \ldots, t-1$ ;
3 **Find closest task:**
   $j^* = \arg\max_j \mathbb{E}_{(x,y)\sim\mathcal{D}^t}[\Delta(\text{NN}(f(x, j|\mathcal{S})), y)]$,
   where NN is the 5-nearest neighbor classifier in feature space;
4 **Define search space:** Take path corresponding to predictor of task $j^*$ and add a new randomly initialized module at every layer. $\Gamma$: distribution over paths; $\pi_i$: $i$-th path;
5 **Split train set in two halves:** $\mathcal{D}_1^t$ and $\mathcal{D}_2^t$;
6 **while** loss in eq. 4 has not converged **do**
7     get sample $(x, y) \sim \mathcal{D}^t[iteration \bmod 2]$;
8     sample path $\pi_k \sim \Gamma$;
9     **if** odd iteration **or** $\max_\Gamma > 0.99$ **then**
10       forward/backward and update $\theta(\pi_k)$ (only newly added modules)
    **else**
11       forward/backward and update $\Gamma$;
    **end**
  **end**
12 Let $i^*$ be the path with largest values in $\Gamma$, then set $f(x, t|\mathcal{S} \cup t)$ to $\pi_{i*}$.

---

**Algorithm 2:** MNTDP-D algorithm.

1 **Data:** Dataset of task $t$: $\mathcal{D}^t$.;
2 **Past predictors:** $f(x, j|\mathcal{S})$ for $j = 1, \ldots, t-1$ ;
3 **Find closest task:**
   $j^* = \arg\max_j \mathbb{E}_{(x,y)\sim\mathcal{D}^t}[\Delta(\text{NN}(f(x, j|\mathcal{S})), y)]$,
   where NN is 5-nearest neighbor classifier in feature space;
4 **Define search space:** Take path corresponding to predictor of task $j^*$ and add a new randomly initialized module at every layer. $\pi_i$: $i$-th path, $i = 1, \ldots, N$ where $N$ is the total number of paths;
5 **for** $i = 1, \ldots, N$ **do**
6     **while** loss in eq. 4 has not converged **do**
7       get sample $(x, y) \sim \mathcal{D}^t$;
8       forward/backward, update parameters $\theta(\pi_i)$ (only newly added modules)
    **end**
9     compute accuracy $A_i$ on validation set.
  **end**
10 Let $i^* = \arg\max_i A_i$, then set $f(x, t|\mathcal{S} \cup t)$ to $\pi_{i*}$.

---

## C  ARCHITECTURE OF THE MODELS AND HYPER-PARAMETERS

The model used on Permuted-MNIST is presented in table 7. On all other tasks, the base architecture for all baselines and MNTDP is the same CNN, a ResNet convolutional neural network as reported in Table 8 or AlexNet as reported in Table 9. The table also shows how layers are grouped into modules for MNTDP and PNN.

| Block | # layers | #params | # hidden units |
|-------|----------|---------|----------------|
| 1 | 1 | 785000 | 1000 |
| 2 | 1 | 1001000 | 1000 |
| 3 | 1 | 10010 | num. classes |

Table 7: Permuted MNIST Model

| Block | # layers | #params | # out channels |
|-------|----------|---------|----------------|
| 1 | 1 | 1856 | 64 |
| 2 | 4 | 147968 | 64 |
| 3 | 4 | 152192 | 64 |
| 4 | 4 | 152192 | 64 |
| 5 | 2 | 78208 | 64 |
| 6 | 2 | 73984 | 64 |
| 7 | 1 | 650 | num. classes |

Table 8: Resnet architecture used throughout our experiments.

| Block | # layers | #params | # out channels/hidden units |
|-------|----------|---------|------------------------------|
| 1 | 1 | 3136 | 64 |
| 2 | 1 | 73856 | 128 |
| 3 | 1 | 131328 | 256 |
| 4 | 1 | 2099200 | 2048 |
| 5 | 1 | 4196352 | 2048 |
| 7 | 1 | 10245 | num. classes |

Table 9: Details of the Alexnet architecture used in Serrà et al. (2018) and in our experiments.

As presented in section 5.2, the stream can be visited only once, preventing stream-level hyper-parameters tuning. Exceptions are made for HAT (Serrà et al., 2018) because we have been using authors' implementation, and for EWC and Online-EWC since these approaches fail in the proposed setting. The constraint strength hyper-parameter $\lambda$ must be tuned at the stream level since a task-level tuning of $\lambda$ results in little or no constraint at all, leading to severe catastrophic forgetting. The stream-level hyper-parameters optimization considers 9 values for $\lambda$ $\{1, 5, 10, 50, 100, 500, 10^3, 5 \times 10^3, 10^4\}$. Note that this gives an unfair advantage to EWC, Online-EWC and HAT, as all other methods including MNTDP use task-level cross-validation as described in §5.2.

For all methods and experiments, we use the Adam optimizer (Kingma & Ba, 2015) with $\beta_1 = 0.9$, $\beta_2 = 0.999$ and $\epsilon = 10^{-8}$.

For each task and each baseline, two learning rates $\{10^{-2}, 10^{-3}\}$ and 3 weight decay strengths $\{0, 10^{-5}, 10^{-4}\}$ are considered. Early stopping is performed on each task to identify the best step to stop training. When the current task validation accuracy stops increasing for 300 iterations, we restore the learner to its state after the best iteration and stop training on the current task.

For MNTDP-S, we consider two additional learning rates for the $\Gamma$ optimization $\{10^{-2}, 10^{-3}\}$. An entropy regularization term on $\Gamma$ is added to the loss to encourage exploration, preventing an early convergence towards a sub-optimal path. The weight for this regularization term is set to 1 throughout our experiments

Finally, since small tasks in $\mathcal{S}^{\text{long}}$ have very few examples in the validation sets, we use test-time augmentation to prevent overfitting during the grid search. For each validation sample, we add four augmented copies following the same data augmentation procedure used during training.

## D    MEMORY COMPLEXITY OF THE MODELS

In the main paper we report the overall memory consumption by the end of training. However, modular architectures like MNTDP use only a sparse subset of modules for a given task. In this section, we report the memory complexity both at training and test time. Table 10 shows that all methods but DEN and RCL are as fast to evaluate at test time as an independent network because they all use a single path.

## E    ADDITIONAL RESULTS

### E.1    METRIC

**LCA**    In addition to the metrics used in the main paper, we report the area under the learning curve after $\beta$ optimization steps (Chaudhry et al., 2019a) as a metric to assess learning speed:

$$\text{LCA@}\beta = \frac{1}{T} \sum_{t=1}^{T} \left[ \frac{1}{\beta + 1} \sum_{b=0}^{\beta} A_{b,t}(f) \right]$$

Where $A_{b,t}(f)$ corresponds to the test set accuracy on task $t$ of learner $f$ after having seen $b$ batches from $t$

|  | Train | Test |
|---|---|---|
| Inde, fintune, new-head | $N$ | $N$ |
| EWC | $N + 2TN$ | $N$ |
| ER | $N + rT$ | $N$ |
| MNTDP-D | $kbN$ | $N$ |
| MNTDP-S | $N + 2kN$ | $N$ |
| HAT | $N$ | $N$ |
| Wide HAT | $SN$ | $SN$ |
| DEN | $N + pT$ | $N + pT$ |
| RCL | $N + pT$ | $N + pT$ |
| Lean to Grow | $N + 2Tp$ | $N$ |

Table 10: Memory complexity of the different baselines at train time and at test time, where $N$ is the size of the backbone, $T$ the number of tasks, $r$ the size of the memory buffer per task, $k$ the number of source columns used by MNTDP, $b$ the number of blocks in the backbone, $S$ the scale factor used for wide-HAT and $p$ the average number of new parameters introduced per task. Note that while Wide HAT and MNTDP-D are using a similar amount of memory on CTrL (Table 1), the inference model used by MNTDP on each task only uses the memory of the narrow backbone, resulting in more than 6 times smaller inference models.

### E.2 CLASSICAL STREAMS

Table 11 and 12 report performance across all axes of evaluation on the standard Permuted MNIST and Split CIFAR 100 streams.

| Model | $< \mathcal{A} >$ | $< \mathcal{F} >$ | Mem. | FLOPs | LCA@5 |
|---|---|---|---|---|---|
| **Independent** | 0.98 | 0.00 | 71.8 | 9.0 | 0.43 |
| **Finetune** | 0.49 | -0.49 | 7.2 | 12.0 | 0.13 |
| **New-head freeze** | 0.89 | 0.00 | 7.5 | 19.0 | 0.14 |
| **New-head finetune** | 0.55 | -0.43 | 7.5 | 15.0 | 0.14 |
| **New-leg freeze** | 0.98 | 0.00 | 35.4 | 10.0 | 0.43 |
| **New-leg finetune** | 0.89 | -0.09 | 35.4 | 14.0 | 0.19 |
| **EWC** † | 0.94 | -0.03 | 143.7 | 18.0 | 0.13 |
| **Online EWC** † | 0.95 | -0.01 | 21.6 | 14.0 | 0.16 |
| **ER (Reservoir)** † | 0.69 | -0.29 | 15.0 | 12.0 | 0.29 |
| **ER** | 0.90 | -0.08 | 15.0 | 15.0 | 0.29 |
| **PNN** | 0.98 | 0.00 | 253.8 | 123.0 | 0.15 |
| **MNTDP-S** | 0.97 | 0.00 | 71.8 | 21.0 | 0.22 |
| **MNTDP-S (k=all)** | 0.98 | 0.00 | 71.8 | 24.0 | 0.19 |
| **MNTDP-D** | 0.98 | 0.00 | 71.8 | 56.0 | 0.37 |
| **HAT**† | 0.95 | 0.00 | 7.6 | 25.0 | 0.11 |
| **HAT (Wide)**† | 0.97 | 0.00 | 78.5 | 209.0 | 0.13 |
| **DEN** † | 0.95 | 0.00 | 8.1 | - | - |
| **RCL** † | 0.96 | 0.00 | 8.5 | 148.2 | - |

Table 11: Results on the standard permuted-MNIST stream. In this stream, each of the 10 tasks corresponds to a random permutation of the input pixels of MNIST digits. For DEN and RCL, since we are using the authors' implementations, we do not have access to the LCA measure. † correspond to models using stream-level cross-validation (see Section 5.2).

| Model | $< \mathcal{A} >$ | $< \mathcal{F} >$ | Mem. | FLOPs | LCA@5 |
|---|---|---|---|---|---|
| **Independent** | 0.82 | 0.00 | 24.4 | 1327.0 | 0.11 |
| **Finetune** | 0.18 | -0.56 | 2.4 | 886.0 | 0.15 |
| **New-head freeze** | 0.57 | 0.00 | 2.5 | 740.0 | 0.21 |
| **New-head finetune** | 0.21 | -0.52 | 2.5 | 687.0 | 0.15 |
| **New-leg freeze** | 0.50 | 0.00 | 2.5 | 1759.0 | 0.12 |
| **New-leg finetune** | 0.18 | -0.42 | 2.5 | 1115.0 | 0.11 |
| **EWC** † | 0.55 | 0.01 | 51.0 | 1151.0 | 0.11 |
| **Online EWC** † | 0.54 | -0.02 | 7.3 | 1053.0 | 0.12 |
| **ER (Reservoir)** † | 0.32 | -0.50 | 20.9 | 1742.0 | 0.17 |
| **ER** | 0.66 | -0.15 | 20.9 | 1524.0 | 0.17 |
| **PNN** | 0.78 | 0.00 | 133.8 | 9889.0 | 0.15 |
| **MNTDP-S** | 0.75 | 0.00 | 24.4 | 1295.0 | 0.11 |
| **MNTDP-S (k=all)** | 0.75 | 0.00 | 24.4 | 1323.0 | 0.11 |
| **MNTDP-D** | 0.83 | 0.00 | 24.3 | 6168.0 | 0.12 |
| **MNTDP-D\*** | 0.81 | 0.00 | 260.9 | 488.0 | 0.17 |
| **HAT\***† | 0.74 | -0.01 | 27.0 | 175.0 | 0.11 |
| **HAT (Wide)\***† | 0.79 | 0.00 | 269.3 | 1830.0 | 0.11 |

Table 12: Results on the standard Split Cifar 100 stream. Each task is composed of 10 new classes.\* correspond to models using an Alexnet backbone, † to models using stream-level cross-validation (see Section 5.2).

### E.3 THE CTRL BENCHMARK

Figure 1 and 6 provide radar plots of the models evaluated on the CTrL benchmark. The companion tables of these plots are in Tab. 13, 14, 15, 16 and 17.

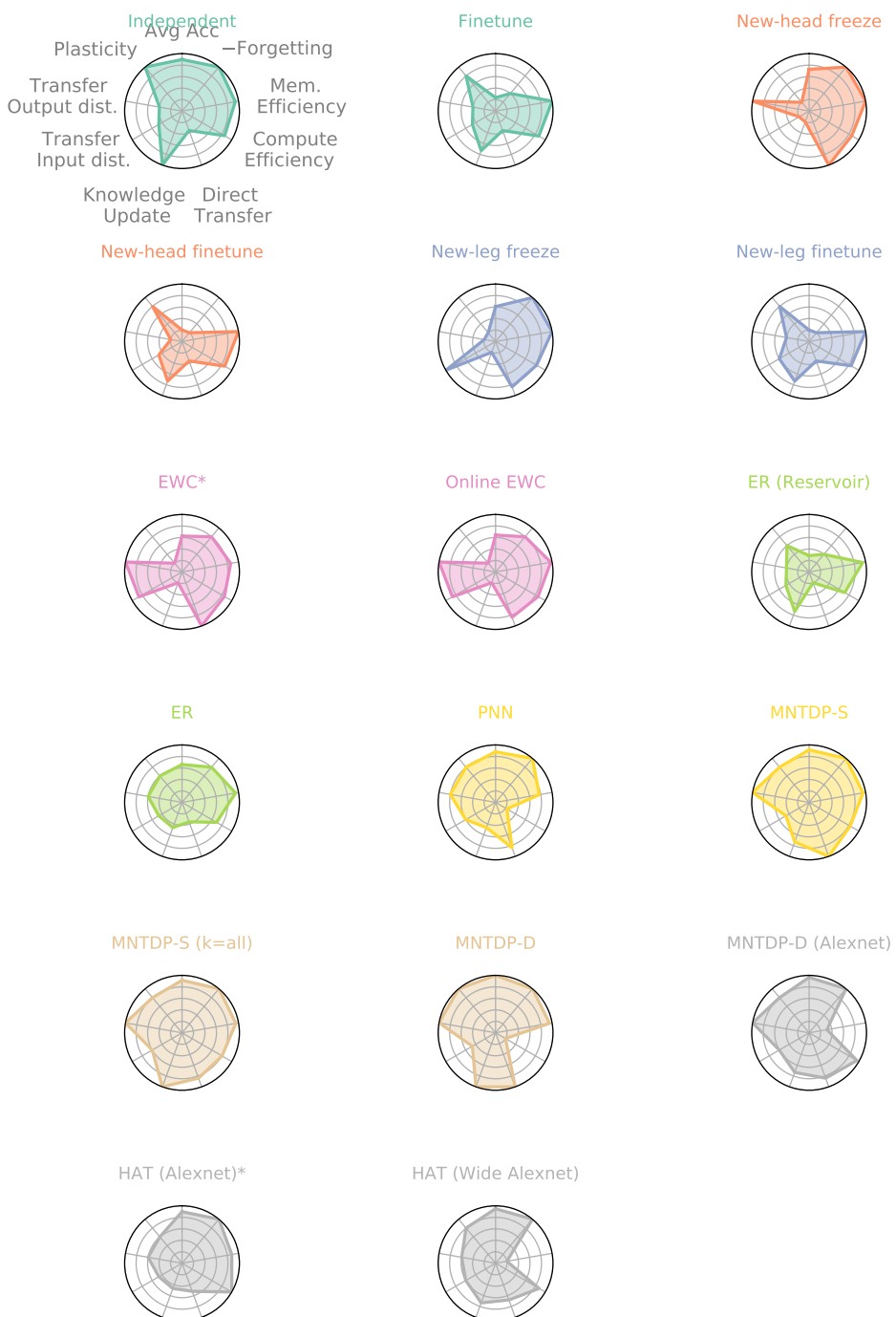

Figure 6: Comparison of the global performance of all baselines on the CTrL Benchmark. MNTDP-D is the most efficient method on multiple of the dimensions, but it requires more computation than MNTDP-S.

E.4   DISCOVERED PATHS

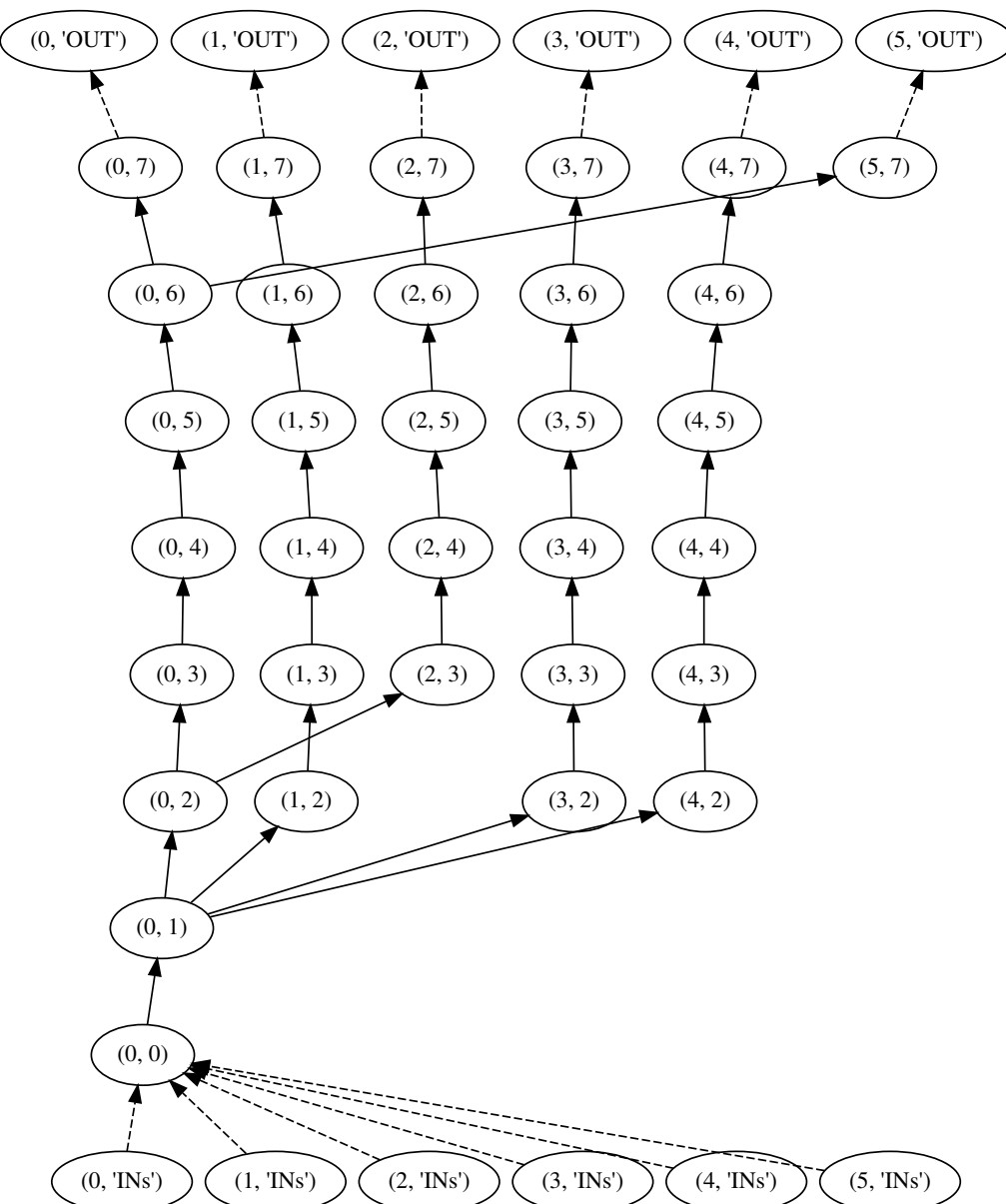

Figure 7: Global graph of paths discovered by MNDTP-D on the $\mathcal{T}(\mathcal{S}^{\text{out}})$ Stream. "INs" (resp. "OUT") nodes are the input (resp. output) of the path for each task. Solid edges correspond to parameterized modules while dashed edges are only used to show which block is selected for each task and don't apply any transformation. We observe that a significant amount of new parameters are introduced for tasks 2, 3, 4 and 5, which are very different from the first task. The model is however able to correctly identify that the last task is very similar to the first one, resulting in very large reuse of past modules and only introducing a new classification layer to adapt to the new task.

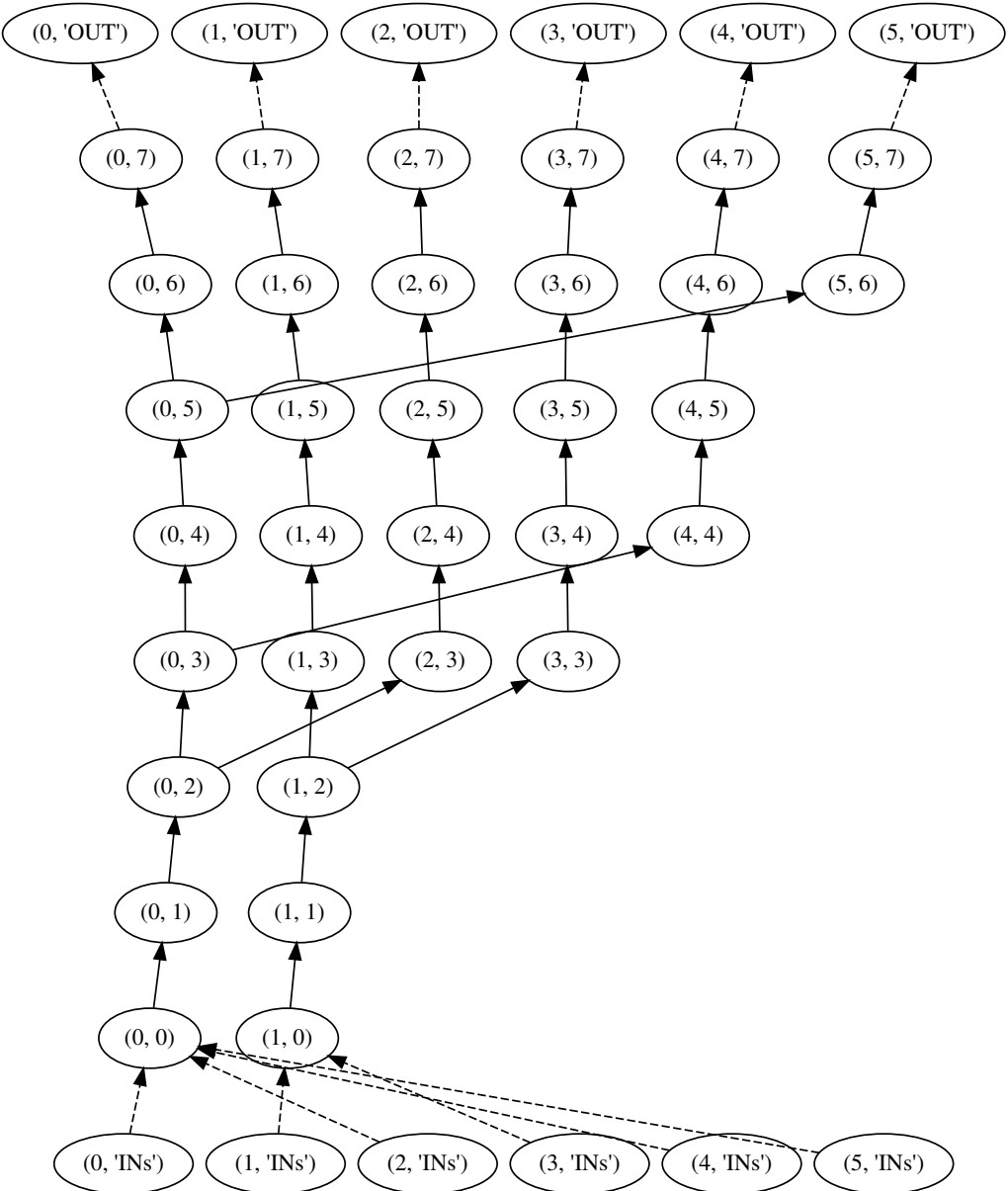

Figure 8: Global graph of paths discovered by MNDTP-S on the $\mathcal{T}(\mathcal{S}^{\text{out}})$ Stream. When facing the last task, the model correctly identified that modules from the first task should be reused, ultimately introducing 2 new modules to solve it.

## E.5 STREAM $\mathcal{S}^-$

| Model | Acc $T_1$ | Acc $T_1'$ | $\Delta_{T_1,T_1'}$ | $\mathcal{T}(\mathcal{S}^-)$ | $<\mathcal{A}>$ | $<\mathcal{F}>$ | Mem. | FLOPs | LCA@5 |
|---|---|---|---|---|---|---|---|---|---|
| **Independent** | 0.72 | 0.36 | -0.35 | 0.00 | 0.56 | 0.00 | 14.6 | 292.0 | 0.10 |
| **Finetune** | 0.72 | 0.37 | -0.34 | 0.01 | 0.18 | -0.33 | 2.4 | 308.0 | 0.10 |
| **New-head freeze** | 0.72 | 0.71 | -0.01 | 0.35 | 0.54 | 0.00 | 2.5 | 262.0 | 0.19 |
| **New-head finetune** | 0.72 | 0.35 | -0.36 | -0.01 | 0.15 | -0.39 | 2.5 | 296.0 | 0.10 |
| **New-leg freeze** | 0.72 | 0.65 | -0.07 | 0.29 | 0.47 | 0.00 | 2.5 | 362.0 | 0.11 |
| **New-leg finetune** | 0.72 | 0.33 | -0.38 | -0.03 | 0.13 | -0.41 | 2.5 | 333.0 | 0.10 |
| **EWC †** | 0.72 | 0.71 | -0.01 | 0.35 | 0.52 | -0.02 | 31.5 | 344.0 | 0.13 |
| **Online EWC †** | 0.72 | 0.69 | -0.03 | 0.33 | 0.54 | -0.01 | 7.3 | 309.0 | 0.11 |
| **ER (Reservoir)†** | 0.72 | 0.30 | -0.41 | -0.06 | 0.20 | -0.32 | 13.5 | 646.0 | 0.12 |
| **ER** | 0.72 | 0.36 | -0.36 | 0.00 | 0.41 | -0.13 | 13.5 | 551.0 | 0.11 |
| **PNN** | 0.72 | 0.65 | -0.06 | 0.29 | 0.62 | 0.00 | 51.1 | 1099.0 | 0.13 |
| **MNTDP-S** | 0.72 | 0.71 | 0.00 | 0.35 | 0.63 | 0.00 | 11.0 | 310.0 | 0.10 |
| **MNTDP-S (k=all)** | 0.72 | 0.63 | -0.09 | 0.27 | 0.61 | 0.00 | 10.7 | 341.0 | 0.10 |
| **MNTDP-D** | 0.72 | 0.72 | 0.00 | 0.36 | 0.67 | 0.00 | 9.2 | 1876.0 | 0.16 |
| **MNTDP-D\*** | 0.64 | 0.64 | 0.00 | 0.28 | 0.63 | 0.00 | 130.2 | 101.0 | 0.21 |
| **HAT\*†** | 0.61 | 0.42 | -0.19 | 0.06 | 0.57 | -0.01 | 26.6 | 54.0 | 0.12 |
| **HAT\*† (Wide)** | 0.67 | 0.54 | -0.13 | 0.18 | 0.60 | 0.00 | 164.0 | 257.0 | 0.14 |

Table 13: Results in the $\mathcal{T}(\mathcal{S}^-)$ evaluation stream. In this stream, the last task is the same as the first with an order of magnitude less data. * correspond to models using an Alexnet backbone, † to models using stream-level cross-validation.

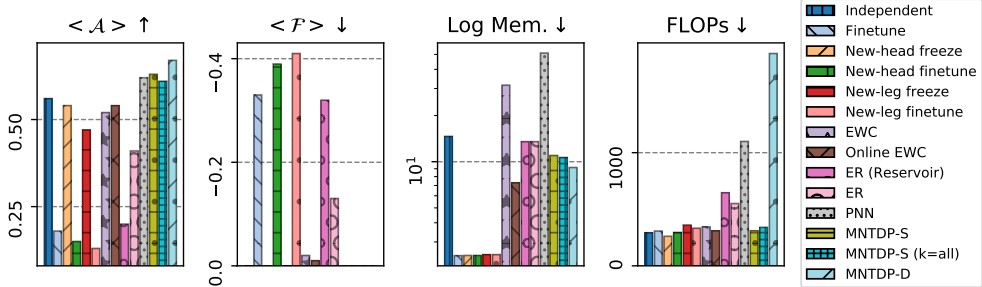

Figure 9: Comparison of all baselines on the $\mathcal{S}^-$ stream.

### E.6 STREAM $\mathcal{S}^+$

| Model | Acc $T_1$ | Acc $T_1'$ | $\Delta_{T_1, T_1'}$ | $\mathcal{T}(\mathcal{S}^+)$ | $<\mathcal{A}>$ | $<\mathcal{F}>$ | Mem. | FLOPs | LCA@5 |
|---|---|---|---|---|---|---|---|---|---|
| **Independent** | 0.37 | 0.71 | 0.35 | 0.00 | 0.57 | 0.00 | 14.6 | 404.0 | 0.10 |
| **Finetune** | 0.37 | 0.58 | 0.21 | -0.13 | 0.24 | -0.23 | 2.4 | 262.0 | 0.12 |
| **New-head freeze** | 0.37 | 0.43 | 0.06 | -0.28 | 0.41 | 0.00 | 2.5 | 361.0 | 0.17 |
| **New-head finetune** | 0.37 | 0.57 | 0.20 | -0.14 | 0.19 | -0.36 | 2.5 | 327.0 | 0.11 |
| **New-leg freeze** | 0.37 | 0.37 | 0.01 | -0.34 | 0.34 | 0.00 | 2.5 | 425.0 | 0.11 |
| **New-leg finetune** | 0.37 | 0.56 | 0.19 | -0.15 | 0.17 | -0.29 | 2.5 | 357.0 | 0.10 |
| **EWC** † | 0.37 | 0.42 | 0.05 | -0.29 | 0.39 | -0.02 | 31.5 | 337.0 | 0.11 |
| **Online EWC** † | 0.37 | 0.40 | 0.03 | -0.31 | 0.37 | -0.04 | 7.3 | 271.0 | 0.11 |
| **ER (Reservoir)** † | 0.37 | 0.56 | 0.20 | -0.15 | 0.20 | -0.33 | 13.5 | 493.0 | 0.11 |
| **ER** | 0.37 | 0.47 | 0.10 | -0.24 | 0.43 | -0.07 | 13.5 | 512.0 | 0.12 |
| **PNN** | 0.37 | 0.54 | 0.17 | -0.17 | 0.52 | 0.00 | 51.1 | 1757.0 | 0.12 |
| **MNTDP-S** | 0.37 | 0.62 | 0.25 | -0.09 | 0.56 | 0.00 | 14.0 | 417.0 | 0.10 |
| **MNTDP-S (k=all)** | 0.37 | 0.66 | 0.29 | -0.05 | 0.56 | 0.00 | 14.0 | 595.0 | 0.10 |
| **MNTDP-D** | 0.37 | 0.72 | 0.35 | 0.01 | 0.61 | 0.00 | 14.0 | 1659.0 | 0.11 |
| **MNTDP-D\*** | 0.40 | 0.64 | 0.24 | -0.07 | 0.61 | 0.00 | 156.3 | 136.0 | 0.17 |
| **HAT\*†** | 0.41 | 0.52 | 0.12 | -0.19 | 0.57 | 0.00 | 26.6 | 39.0 | 0.13 |
| **HAT\*† (Wide)** | 0.41 | 0.61 | 0.19 | -0.10 | 0.59 | 0.00 | 164.0 | 324.0 | 0.14 |

Table 14: Results in the $\mathcal{S}^+$ evaluation stream. In this stream, the 5th task is the same as the first with an order of magnitude more data. Tasks 2, 3, and 4 are distractors. * correspond to models using an Alexnet backbone, † to models using stream-level cross-validation.

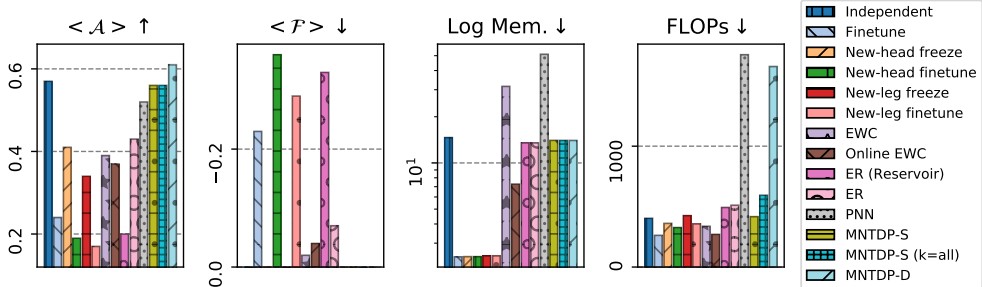

Figure 10: Comparison of all baselines on the $\mathcal{S}^+$ stream

### E.7 STREAM $\mathcal{S}^{\text{IN}}$

| Model | Acc $T_1$ | Acc $T_1'$ | $\Delta_{T_1,T_1'}$ | $\mathcal{T}(\mathcal{S}^{\text{in}})$ | $<\mathcal{A}>$ | $<\mathcal{F}>$ | Mem. | FLOPs | LCA@5 |
|---|---|---|---|---|---|---|---|---|---|
| **Independent** | 0.98 | 0.60 | -0.39 | 0.00 | 0.57 | 0.00 | 14.6 | 265.0 | 0.10 |
| **Finetune** | 0.98 | 0.57 | -0.41 | -0.03 | 0.18 | -0.31 | 2.4 | 244.0 | 0.11 |
| **New-head freeze** | 0.98 | 0.39 | -0.59 | -0.21 | 0.45 | 0.00 | 2.5 | 246.0 | 0.13 |
| **New-head finetune** | 0.98 | 0.62 | -0.36 | 0.02 | 0.19 | -0.33 | 2.5 | 188.0 | 0.11 |
| **New-leg freeze** | 0.98 | 0.95 | -0.04 | 0.35 | 0.48 | 0.00 | 2.5 | 282.0 | 0.10 |
| **New-leg finetune** | 0.98 | 0.70 | -0.28 | 0.10 | 0.21 | -0.34 | 2.5 | 238.0 | 0.10 |
| **EWC †** | 0.98 | 0.87 | -0.12 | 0.27 | 0.43 | -0.14 | 31.5 | 269.5 | 0.11 |
| **Online EWC †** | 0.98 | 0.87 | -0.12 | 0.27 | 0.43 | -0.12 | 7.3 | 287.0 | 0.10 |
| **ER (Reservoir) †** | 0.98 | 0.60 | -0.38 | 0.00 | 0.30 | -0.26 | 13.5 | 570.0 | 0.12 |
| **ER** | 0.98 | 0.60 | -0.38 | 0.00 | 0.38 | -0.17 | 13.5 | 604.0 | 0.12 |
| **PNN** | 0.98 | 0.70 | -0.29 | 0.10 | 0.57 | 0.00 | 51.1 | 899.0 | 0.10 |
| **MNTDP-S** | 0.98 | 0.64 | -0.34 | 0.04 | 0.57 | 0.00 | 12.2 | 333.0 | 0.10 |
| **MNTDP-S (k=all)** | 0.98 | 0.68 | -0.30 | 0.08 | 0.57 | 0.00 | 13.4 | 327.0 | 0.10 |
| **MNTDP-D** | 0.98 | 0.62 | -0.36 | 0.02 | 0.60 | 0.00 | 11.6 | 1225.0 | 0.11 |
| **MNTDP-D\*** | 0.98 | 0.67 | -0.31 | 0.07 | 0.59 | 0.00 | 156.6 | 115.0 | 0.12 |
| **HAT\*†** | 0.98 | 0.61 | -0.37 | 0.01 | 0.58 | -0.01 | 26.6 | 41.0 | 0.12 |
| **HAT\*† (Wide)** | 0.97 | 0.67 | -0.30 | 0.07 | 0.62 | 0.00 | 164.0 | 186.0 | 0.11 |

Table 15: Results in the transfer evaluation stream with input perturbation. In this stream, the last task is the same as the first one with a modification applied to the input space and with an order of magnitude less data. Tasks 2, 3, 4 and 5 are distractors. * correspond to models using an Alexnet backbone, † to models using stream-level cross-validation.

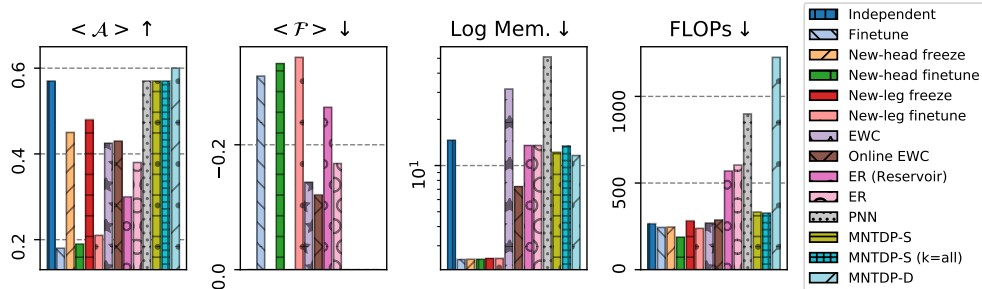

Figure 11: Comparison of all baselines on the $\mathcal{S}^{\text{in}}$ stream

## E.8 STREAM $\mathcal{S}^{\text{OUT}}$

| Model | Acc $T_1$ | Acc $T_1'$ | $\Delta_{T_1,T_1'}$ | $\mathcal{T}(\mathcal{S}^{\text{out}})$ | $<\mathcal{A}>$ | $<\mathcal{F}>$ | Mem. | FLOPs | LCA@5 |
|---|---|---|---|---|---|---|---|---|---|
| **Independent** | 0.70 | 0.37 | -0.32 | 0.00 | 0.61 | 0.00 | 14.6 | 349.0 | 0.10 |
| **Finetune** | 0.70 | 0.36 | -0.33 | -0.01 | 0.15 | -0.37 | 2.4 | 331.0 | 0.11 |
| **New-head freeze** | 0.70 | 0.70 | 0.01 | 0.33 | 0.54 | 0.00 | 2.5 | 374.0 | 0.19 |
| **New-head finetune** | 0.70 | 0.31 | -0.39 | -0.06 | 0.14 | -0.41 | 2.5 | 379.0 | 0.10 |
| **New-leg freeze** | 0.70 | 0.25 | -0.45 | -0.12 | 0.40 | 0.00 | 2.5 | 369.0 | 0.10 |
| **New-leg finetune** | 0.70 | 0.33 | -0.36 | -0.04 | 0.14 | -0.40 | 2.5 | 340.0 | 0.10 |
| **EWC †** | 0.70 | 0.68 | -0.01 | 0.31 | 0.52 | -0.03 | 31.5 | 387.0 | 0.11 |
| **Online EWC †** | 0.70 | 0.66 | -0.04 | 0.29 | 0.51 | -0.03 | 7.3 | 399.0 | 0.10 |
| **ER (Reservoir) †** | 0.70 | 0.39 | -0.31 | 0.02 | 0.22 | -0.35 | 13.5 | 732.0 | 0.11 |
| **ER** | 0.70 | 0.50 | -0.19 | 0.13 | 0.54 | -0.07 | 13.5 | 758.0 | 0.11 |
| **PNN** | 0.70 | 0.62 | -0.07 | 0.25 | 0.62 | 0.00 | 51.1 | 1799.0 | 0.10 |
| **MNTDP-S** | 0.70 | 0.64 | -0.05 | 0.27 | 0.64 | 0.00 | 10.1 | 406.0 | 0.11 |
| **MNTDP-S (k=all)** | 0.70 | 0.63 | -0.06 | 0.26 | 0.63 | 0.00 | 11.6 | 411.0 | 0.10 |
| **MNTDP-D** | 0.70 | 0.70 | 0.01 | 0.33 | 0.68 | 0.00 | 11.6 | 1299.0 | 0.15 |
| **MNTDP-D\*** | 0.64 | 0.64 | 0.00 | 0.27 | 0.65 | 0.00 | 130.2 | 99.0 | 0.22 |
| **HAT\*†** | 0.62 | 0.44 | -0.18 | 0.07 | 0.60 | 0.00 | 26.6 | 42.0 | 0.13 |
| **HAT\*†** | 0.68 | 0.51 | -0.17 | 0.14 | 0.64 | 0.00 | 164.0 | 293.0 | 0.12 |

Table 16: Results in the transfer evaluation stream with output perturbation. In this stream, the last task uses the same classes as the first task but in a different order and with an order of magnitude less data. Tasks 2, 3, 4 and 5 are distractors. \* correspond to models using an Alexnet backbone, † to models using stream-level cross-validation.

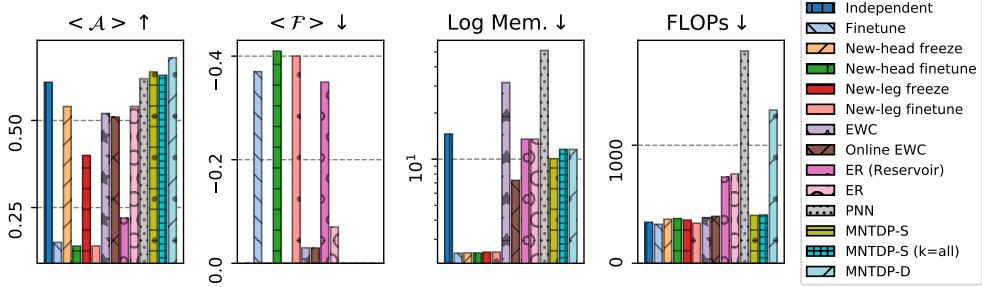

Figure 12: Comparison of all baselines on the $\mathcal{S}^{\text{out}}$ stream

E.9 $\mathcal{S}^{\text{PL}}$

| Model | Acc $T_5$ | $\Delta_{T_5',T_5}$ | $<\mathcal{A}>$ | $<\mathcal{F}>$ | Mem. | FLOPs | LCA@5 |
|---|---|---|---|---|---|---|---|
| **Independent** | 0.71 | 0.00 | 0.59 | 0.00 | 12.2 | 232.0 | 0.10 |
| **Finetune** | 0.57 | -0.14 | 0.21 | -0.30 | 2.4 | 274.0 | 0.12 |
| **New-head freeze** | 0.29 | -0.42 | 0.45 | 0.00 | 2.4 | 294.0 | 0.13 |
| **New-head finetune** | 0.56 | -0.15 | 0.20 | -0.35 | 2.4 | 336.0 | 0.11 |
| **New-leg freeze** | 0.27 | -0.44 | 0.37 | 0.00 | 2.5 | 390.0 | 0.11 |
| **New-leg finetune** | 0.58 | -0.13 | 0.19 | -0.34 | 2.5 | 375.0 | 0.11 |
| **EWC †** | 0.28 | -0.43 | 0.27 | -0.19 | 26.7 | 239.0 | 0.11 |
| **Online EWC †** | 0.28 | -0.43 | 0.30 | -0.17 | 7.3 | 282.0 | 0.12 |
| **ER (Reservoir)†** | 0.48 | -0.23 | 0.20 | -0.27 | 11.7 | 383.0 | 0.10 |
| **ER** | 0.51 | -0.20 | 0.45 | -0.08 | 11.7 | 597.0 | 0.10 |
| **PNN** | 0.56 | -0.15 | 0.54 | 0.00 | 36.5 | 1742.0 | 0.13 |
| **MNTDP-S** | 0.58 | -0.13 | 0.55 | 0.00 | 11.0 | 351.0 | 0.10 |
| **MNTDP-S (k=all)** | 0.60 | -0.11 | 0.56 | 0.00 | 11.0 | 340.0 | 0.11 |
| **MNTDP-D** | 0.70 | -0.01 | 0.62 | 0.00 | 11.6 | 1503.0 | 0.10 |
| **MNTDP-D\*** | 0.65 | -0.06 | 0.64 | 0.00 | 130.2 | 124.0 | 0.17 |
| **HAT\*†** | 0.50 | -0.21 | 0.58 | 0.00 | 26.5 | 50.0 | 0.11 |
| **HAT\*† (Wide)** | 0.61 | -0.10 | 0.61 | 0.00 | 163.7 | 312.0 | 0.12 |

Table 17: Results in the plasticity evaluation stream. In this stream, we compare the performance on the probe task when it is the first problem encountered by the model and when it as already seen 4 distractor tasks.* correspond to models using an Alexnet backbone, † to models using stream-level cross-validation (see Section 5.2)

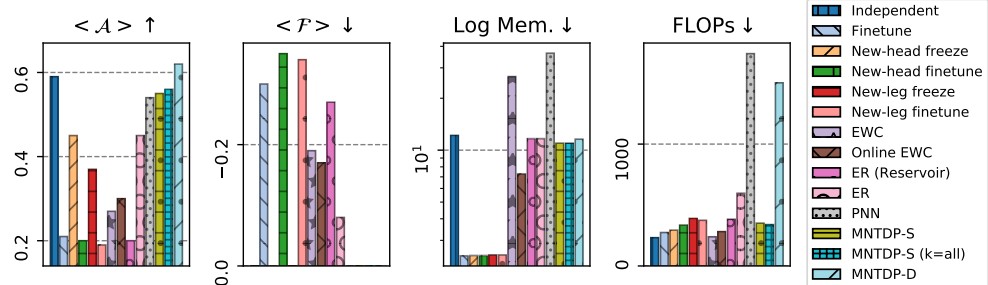

Figure 13: Comparison of all baselines on the $\mathcal{S}^{\text{pl}}$ stream

## E.10 $\mathcal{S}^{\text{LONG}}$

| Model | $<\mathcal{A}>$ | $<\mathcal{F}>$ | Mem. | FLOPs | LCA@5 |
|---|---|---|---|---|---|
| **Independent** | $0.57 \pm 0.01$ | $0.0 \pm 0.0$ | $243.7 \pm 0.0$ | $3542.33 \pm 139.16$ | $0.2 \pm 0.0$ |
| **Finetune** | $0.2 \pm 0.0$ | $-0.35 \pm 0.0$ | $2.4 \pm 0.0$ | $4961.33 \pm 112.16$ | $0.23 \pm 0.0$ |
| **New-head freeze** | $0.43 \pm 0.01$ | $0.0 \pm 0.0$ | $2.6 \pm 0.0$ | $5574.33 \pm 249.65$ | $0.27 \pm 0.0$ |
| **Online EWC** † | $0.27 \pm 0.01$ | $-0.25 \pm 0.01$ | $7.4 \pm 0.0$ | $3882.67 \pm 159.15$ | $0.21 \pm 0.0$ |
| **MNTDP-S** | $0.68 \pm 0.0$ | $0.0 \pm 0.0$ | $158.63 \pm 2.58$ | $5437.67 \pm 110.77$ | $0.21 \pm 0.0$ |
| **MNTDP-D** | $0.75 \pm 0.0$ | $0.0 \pm 0.0$ | $102.03 \pm 0.8$ | $26066.67 \pm 662.74$ | $0.34 \pm 0.0$ |
| **MNTDP-D*** | $0.75 \pm 0.0$ | $0.0 \pm 0.0$ | $1803.47 \pm 16.45$ | $2598.67 \pm 70.48$ | $0.46 \pm 0.0$ |
| **HAT*†** | $0.24 \pm 0.01$ | $-0.1 \pm 0.03$ | $31.9 \pm 0.0$ | $147.0 \pm 27.39$ | $0.21 \pm 0.0$ |
| **HAT (Wide) *†** | $0.32 \pm 0.0$ | $0.0 \pm 0.0$ | $285.0 \pm 0.0$ | $1056.33 \pm 137.29$ | $0.21 \pm 0.0$ |

Table 18: Results on the long evaluation stream. We report the mean and standard error using 3 different instances of the stream, all generated following the procedure described in A. * correspond to models using an Alexnet backbone, † to models using stream-level cross-validation (see Section 5.2)

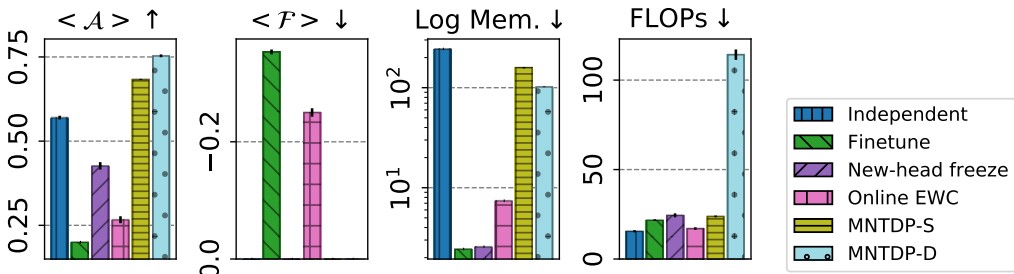

Figure 14: Comparison of all baselines on the $\mathcal{S}^{\text{long}}$ stream

