# OpenReview forum: "Efficient Continual Learning with Modular Networks and Task-Driven Priors"
_ICLR.cc/2021/Conference — ICLR 2021 Poster_

### Official Review · AnonReviewer3 · 2020-10-28
**Continual learning by expanding modular networks**

**Rating:** 7
**Confidence:** 3

**Review:**

The authors investigate modular neural networks in a continual learning (CL) setting, where multiple classification tasks have to be learned in sequence. To avoid the well-known problem of catastrophic forgetting, parameters used by previous tasks are simply frozen, and new tasks are learned by expanding the architecture (as in e.g. PNN, Rusu et al., 2016). The authors focus on efficient forward transfer to new tasks. A many-tasks benchmark is evaluated on the authors' method as well as a number of baselines.

The paper is very well written and the explanations are clear. The experiments are fair and well-executed. The proposed long-sequence streams provide useful baselines that can be used in future papers, as the authors will share their code.

The main weaknesses of the paper lie in its originality and applicability. Dynamic architecture expansion for CL is a well-studied approach (as acknowledged by the authors), and while picking an appropriate initial path with a cheap classifier is a good idea, determining how to expand from there remains difficult and largely expensive in the more effective MNTDP-D version.

Here are some suggested improvements for the authors:
- Baselines: since PNN, improved methods for architecture expansion have been developed. While the provided baselines are interesting and evaluated in a fair manner, I think that the paper would be much stronger if the authors considered as well stronger, more comparable baselines (for example, DEN, Yoon et al., 2018, or CLAW, Adel et al., 2020).

- Other methods have been developed which leverage reinforcement learning to help determine how to expand a network in a CL setting. These should be at least discussed in the light of MNTDP-S (e.g., Xu and Zhu, NeurIPS 2018 and NHuang, Lavet, Rabusseau, arXiv 2019)

- The proposed method cannot achieve backward transfer. This should be noted.

- It should be discussed that the presented method requires knowledge of task identity at test time. This is not always possible.

- "Finally, the CL algorithm has to yield predictors that scale sublinearly with the number of tasks both in terms of memory and compute."
To make the statement general, I would note that sublinear memory and time complexity scaling with the number of tasks should be the aim whenever the continual learner is exposed to tasks that are not independent.

Edit: I have raised my score from 6 to 7 after the rebuttal.

---

> ### Author Response · Authors · 2020-11-21
> **Thanks for your comments and remarks. We have submitted a new version of the paper with many improvements as explained in the comment addressed to all the reviewers. We provide detailed answers to your specific questions:**
>
> - “while picking an appropriate initial path with a cheap classifier is a good idea, determining how to expand from there remains difficult and largely expensive in the more effective MNTDP-D version.” It is true that the space of possibilities is unexplored, but this work provides a very simple yet powerful baseline, and it is therefore a quite valuable starting point. MNTDP-D can be more or less expensive depending on how much the seed architecture(s) are perturbed; this can be adjusted depending on the computational budget. In our experiments, we only need to train 6 networks in parallel, a budget that is constant w.r.t. number of tasks, unlike other approaches. Moreover, the cost at test time is the same as independent nets, as each task is eventually associated to a single path.
> - “Compare to for example, DEN, Yoon et al., 2018, or CLAW, Adel et al., 2020”: We have added comparison to DEN (Fig. 3 and Appendix E), only for MNIST since the provided code only supports fully connected nets. We could not compare to CLAW because the code is not publicly available yet. We have added comparisons to HAT (Serra et al. 2018) on MNIST, CIFAR100 and CTrL (Fig. 3, 4, Tab. 1 and 2), and we have also added a comparison to RCL (Xu et al. 2018) on MNIST (Fig. 3 and Appendix E). On the shorter and simpler streams, these methods work similarly (although MNTDP attains higher average accuracy), on the longer stream MNTDP works much better compared to HAT. The conclusion is that methods that do not restrict the search space (like RCL, Xu et al. 2018) cannot scale effectively to S^long.
> - “their methods have been developed which leverage reinforcement learning to help…”: Thank you for these references. We have discussed Xu et al. 2018 in Section 2 now, and compared to it in Fig. 3 and Appendix E. Huang, Lavet, Rabusseau (arXiv 2019)  consider a different setting where, at stage t, all the datasets of the previous task are still available while we assume access only to the dataset of the current task.
> - “The proposed method cannot achieve backward transfer. This should be noted.”: Yes, we state  “Note that only the parameters of the newly added modules are subject to training, de facto preventing forgetting of previous tasks by construction but also preventing positive backward transfer.” (sec. 4.1).
> - “It should be discussed that the presented method requires knowledge of task identity at test time. This is not always possible.”: Yes, we state: “Task descriptors are provided to the learner both during training and test time.” (first paragraph of sec. 3), and we define our predictor as f(x,t|S), which depends on the task descriptor.
> - "Finally, the CL algorithm has to yield predictors that scale sublinearly with the number of tasks both in terms of memory and compute." To make the statement general, I would note that sublinear memory and time complexity scaling with the number of tasks should be the aim whenever the continual learner is exposed to tasks that are not independent.”: Yes, we contextualized this as suggested. For instance, we state in the second paragraph of the introduction “the learner should be able to scale sub-linearly with the number of tasks, both in terms of memory and compute, when these are related.”.
>
> Thank you.

---

> > ### Comment · AnonReviewer3 · 2020-11-23
> > **Score updated**
> >
> > Thank you for the revision. I think that the paper improved with the new experiments and changes to the text, in particular the comparison to HAT-wide made the paper stronger. I have updated my score from a 6 to a 7.
> >
> > I encourage the authors to make their benchmarks publicly available and easy to use.

---

### Official Review · AnonReviewer2 · 2020-10-28
**AnonReviewer2 Review**

**Rating:** 6
**Confidence:** 4

**Review:**

**Summary of paper**

This paper has two main contributions. Firstly, it introduces a new set of benchmarks designed to proble different qualities that may be important for continual learning methods. Secondly, it introduces a new modular method for continual learning, which learns to add modules when learning on a new task. The method reduces the exponential search-space by only considering a few options for how to add the module, leveraging which task the new task is most similar to. Experimental results are provided on the new benchmarks as well as Permuted MNIST and Split CIFAR100.

**Review summary**

I really like the idea behind the benchmark suite for continual learning. Although the algorithm CTrL is neat, I think the experiments/explanation could do with more work (see later in the review). Therefore I am currently leaning towards a weak reject, pending a discussion with the authors.

**Pros of paper**

1. Testing different properties of a continual learning algorithm, one by one, with different benchmarks, is a very nice idea. In fact, it might be nice for the authors to consider expanding the suite of benchmarks to test even more desirable aspects of continual learning / properties of continual learning methods. Some ideas:
(i) Multiple tasks which are from the same dataset, to test how continual learning systems react. Some will deal very well with this, others will not.
2. It's great to see memory and compute reported as benchmarks for continual learning.
3. The proposed modular method is very nicely presented, and is a neat, relatively simple, method. It has nice ideas to it, such as starting with the modular architecture of the most related task only.

**Cons of paper**

4. The paper defines a transfer metric, using only the final task. This works well with the specific benchmarks they propose. This is very similar to a "Forward Transfer Metric" from Pan et al., 2020 ("Deep Continual Learning by Functional Regularisation of Memorable Past"), which averages the metric over all tasks (except the first task). I was wondering why the authors restricted the transfer metric to be just for the last task.
5. Although the related work section is long with some nice detail in places, there could be more comparisons in the paper to model-based continual learning methods. The paper could also mention methods based on IBP priors.
6. I would like more of a discussion relating to Li et al., 2019, which the authors say is the most related work. It would have been nice to use that as a baseline in the experiments. (Although I understand it is difficult if they do not have code available online.) In general, I would have liked to see some stronger / more recent baselines in the experiment section: Independent and PNNs are known to have bad memory growth; ER is the simplest idea in the replay-based methods; Online EWC is also very simple (and old).
7. I did not understand why the "Independent" baseline in Table 1 has non-zero transfer metrics? (I thought that, because it is independent models, the metric should be 0?)

**Update to review**

I have increased the score from 5 to 6. The authors have made the paper stronger with the inclusion of stronger baselines, cleaned-up presentation, and backward transfer. I am particularly excited by the new suite of benchmarks.

---

> ### Author Response · Authors · 2020-11-21
> **We thank you for your review. We have submitted a new version of the paper (see comment to all reviewers). We provide detailed answers to the specific points you raised.**
>
> - “it might be nice for the authors to consider expanding the suite of benchmarks”: we  agree (see also answer to reviewer 1). One of the objectives of this work is to provide a benchmark evaluating CL models on multiple dimensions. We completely agree that other dimensions should be measured, such as the ability to deal with different nature of data, with compositional task descriptors, with noisy data, etc… This is a first step and we hope to extend the benchmark in the future. In the paper, we now say “Other dimensions of transfer (e.g.,  transfer with compositional task descriptors or under noisy conditions) are avenue of future work.“ at the end of section 3. However, the current version already provides a finer-grained evaluation of CL models which is a useful contribution. If you look at the newly added Fig. 1, you can appreciate how easy it is now to assess strengths and weaknesses of various methods along these basic axes.
> - “This is very similar to a "Forward Transfer Metric" from Pan et al., 2020 ("Deep Continual Learning by Functional Regularisation of Memorable Past"), which averages the metric over all tasks (except the first task). I was wondering why the authors restricted the transfer metric to be just for the last task.” The reason why we do not average is that, by construction, only the last task relates in some ways to the first task, the other tasks are rather unrelated. Therefore, we do not expect any model to be able to do any better than an independent model on the intermediate tasks, but instead focus the measurement on the last task because we aim at assessing transfer from the very first task (for S^-, S^+, S^in and S^out). Reporting the metric in Pan et al. is certainly interesting for the long stream.
> - “more comparisons in the paper to model-based continual learning methods. The paper could also mention methods based on IBP priors.”: We were not aware of the work by Kessler et al. 2019 on CL with Indian Buffet Processes, thank you for pointing it out. We have updated the related work section accordingly (section 2). In general, approaches  which are based on gating of individual hidden units are difficult to scale to a large number of tasks because the random pattern of sparsity of the hidden state cannot be efficiently  leveraged by GPU devices. The reason why MNTDP scales is because at any given time, both at training and test time, we operate on a small subset of modules (whose number is constant w.r.t. number of tasks).  Concerning model-based continual learning methods, you may be referring to continual learning in reinforcement learning while we focus on supervised learning. Do you have references in the supervised learning setting?
> - “I would have liked to see some stronger / more recent baselines in the experiment section”: Li et al do not provide code to reproduce their work, unfortunately. We have added the HAT and HAT-wide baselines as per suggestion of R4 and yours (see Fig. 3 and 4 and Tab. 1 and 2 in Section 5 of the updated paper). On the permuted MNIST, we also added the RCL and DEN models which provide implementations for MNIST-like datasets (Figure 3 and Appendix E)
> - “I did not understand why the "Independent" baseline in Table 1 has non-zero transfer metrics?”: thank you for pointing this out, it was a mistake. All tables have now been updated with the correct values (which are indeed 0 for Independent Nets); conclusions remain the same.
>
> Please, let us know if we have adequately addressed your concerns. Thank you!

---

> > ### Comment · AnonReviewer2 · 2020-11-23
> > **Re**
> >
> > Thanks to the authors for their response. The paper has become stronger with the inclusion of additional baselines.
> >
> > - Re forward transfer metric: as the authors say, only the last task is used in this metric because, by construction, only the last task is related to the first. Although this is not very general, I think this is fine as a (suite of) benchmarks: it's certainly better than many other attempts in the field right now. However, considering that the CTrL benchmarks are supposed to be general, why is the backward transfer metric over many tasks (in contrast to the forward transfer metric)? I understand that the MNTDP method has no backward transfer, but perhaps it is still worth spending some more time on benchmarking backward transfer for other algorithms.
> > - I am in general much more excited by the new suite of benchmarks than the new method MNTDP, which seems like a smaller change on previous methods. Therefore I am suggesting the authors focus the paper writing to highlight the benchmark (and many algorithms' evaluation on it) more.
> > - Question: in the worst case, if the most similar task to the current task is always the most recent task, does the algorithm memory size grow linearly with task?

---

> > > ### Author Response · Authors · 2020-11-23
> > > **Re**
> > >
> > > Thank you for your response.
> > > - We can definitely add backward transfer in our next revision. We provide here the backward transfer measured as $BWT(\mathcal{S})= \mathbb{E}_{(x,y) \sim \mathcal{D}^1}  [\Delta(f(x, 1 |  \mathcal{S} = 1, \dots, T), y) - \Delta(f(x, 1 |  \mathcal{S}' = 1), y)$ on $S^+$ (which is the first term of the BWT defined by Lopez-Paz et al 2017), the best suited stream for measuring it on our benchmark. We observe that backward transfer is absent from all methods but finetune, showing that there is room for improvement in that dimension. Thank you for the suggestion.
> > >
> > >      |             |   $BWT(S^+)$ |
> > >      |-------------|------:|
> > >      | Independent |   0.0 |
> > >      | Finetune    |  0.21 |
> > >      | New-head    |   0.0 |
> > >      | New-leg     |   0.0 |
> > >      | Online EWC  | -0.05 |
> > >      | ER          | -0.03 |
> > >      | PNN         |   0.0 |
> > >      | MNTDP-S     |   0.0 |
> > >      | MNTDP-D     |   0.0 |
> > >      | MNTDP-D*    |   0.0 |
> > >      | HAT*        | -0.01 |
> > >      | HAT (Wide)* |   0.0 |
> > >
> > > - If you try the approaches we compared against on $S^{long}$, you will find that none of them works (they either do not scale or they do not work well). Isn’t this surprising? The idea of the data driven prior might appear very simple, but it is very effective. Therefore, we believe it is important for the community to know about it. At the very least, MNTDP should serve as a strong baseline of comparison for more sophisticated approaches. We are ready to release a repository just with a data provider for CTrL. Separately, we have another repository for MNTDP evaluated on CTrL and the other benchmarks. This should make it clear that the value of CTrL is disjoint from a particular model, although MNTDP shows the strongest performance on it at present.
> > > - No, the worst case scenario is when none of the previous tasks is related to the current task (relative to the amount of data available to train on the current task). In this case MNTDP will add a new module at every layer and it does not matter which tasks is deemed the most similar one (it does not need to be the last one), MNTDP will still add a new module at every layer if sharing does not yield superior accuracy on the validation set.

---

### Official Review · AnonReviewer4 · 2020-10-29
**Modularized network for continual learning**

**Rating:** 6
**Confidence:** 4

**Review:**

This paper proposes modular networks with task driven prior (MNTNP) for continual learning. For each new task, the model will try to add new modules to each layer, and then search for a best path that leads to the best performance for the current task. During this process only the newly added modules are tuned, therefore, the method does not have catastrophic forgetting by construction. The model size is likely to grow through this process. However, due to similarity among tasks, it is likely that some of the modules will be shared between different tasks. Hence, the growth of the size of the model is likely to be sub-linear with respect to the number of tasks.

+ Pros
+ The idea of exploring different modules and structures for each task is interesting

+ The paper is in general well written and easy to follow

- Cons
- The novelty of the proposed method is limited. The only difference it has as compared to [1] seems to be the data prior that it used to get the previous path to start the search.  There is no comparison with [1] or the other similar methods such as [2], and [3].  It is hard to see if the proposed change resulted in a significant benefit for the learning. In addition, the comparison with other methods are not complete. Because some of the method does not add additional parameter for each of the task, it is hard to see if the gain from the proposed method or is more of a matter of adding additional parameters to the model.

- The setting to freeze old path does prevent forgetting by construction, however, it also prevents positive backward transfer as a whole. This takes away half of the benefit from continual learning, and the only thing left is forward transfer. This scheme also limits forward transfer. For example, in the $S^+$ setting, in an ideal case the model can only tune the final classification layer according to this method and lose the chance to improve the overall model. The authors argue that the optimal model for permuted MNIST and split CIFAR are independent models because each task is distinct. I would argue that, for permuted MNIST the semantic information is consistent across tasks, so ideally a continual learning model would only need to re-arrange the initial layers to accommodate the change in surface form and keep the rest of the model intact. For CIFAR the sharing would be more significant, because the input were mostly drawn from the same distribution and it is likely to have sharable information across tasks. In fact, I would argue the split to independent model is no longer continual learning, as the models are already independent, we completely lost the continual part.

- The proposed approach also requires a task identifier to work with. I think in classification setting, which is the experiments focused in the paper, providing this identifier alone already solves a large part of the problem. The setting is more realistic in reinforcement learning setting, however, no experiments were carried out under that setting.

- It is not clear what algorithm is used in the experiments for the proposed method. Is the data driven prior always applied?


References:
[1] Li, X., Zhou, Y., Wu, T., Socher, R., & Xiong, C. (2019, May). Learn to Grow: A Continual Structure Learning Framework for Overcoming Catastrophic Forgetting. In International Conference on Machine Learning
[2] Serra, J., Suris, D., Miron, M., & Karatzoglou, A. (2018, July). Overcoming Catastrophic Forgetting with Hard Attention to the Task. In International Conference on Machine Learning
[3] Yoon, J., Yang, E., Lee, J., & Hwang, S. J. (2018, February). Lifelong Learning with Dynamically Expandable Networks. In International Conference on Learning Representations.

---

> ### Author Response · Authors · 2020-11-21
> **We thank you for your remarks. We have submitted a new version of the paper (see comment to all reviewers) which should have addressed your main concerns. (1/2)**
>
> We would like to emphasize that the CTrL benchmark is an important contribution of the paper in itself. We are open-sourcing it as a standalone once the anonymity period is over. It enables evaluation across  multiple dimensions of transfer in the continual learning setting and also it enables evaluation on a long stream of (non-trivial) tasks. Arguably, the literature is missing standardized benchmarks and metrics allowing a fair comparison of methods, and we expect our benchmark to be a first step towards better reproducibility in the continual learning setting (see also our answer to Reviewer 1). We have updated the paper (see Section 1) to make our contributions clearer.
>
> Let us now answer each of the specific points you raised:
> - “The novelty of the proposed method is limited”: the novelty in terms of modeling lies in the data-driven prior (see updated Section 4.2) that we use not only to seed the search but also to restrict the search space which otherwise becomes unfeasibly too large when the number of tasks and modules increase over time. It is a simple idea which could be applied to other methods as well, like RCL (Xu et al. 2018) for instance. We certainly do not claim novelty for the use of a modular architecture which has clearly been done before. In the related section we now clarify our contribution as follows: “These two works [Li et al. 2019 and Xu et al. 2018] are the most similar to ours, with the major difference that we restrict the search space over architectures, enabling much better scaling to longer streams.  While their search space (and RAM consumption) grows over time, ours is constant. Our approach is modular , and only a small (and constant) number of modules is employed for any given task both at training and test time.”. Please, see the revised related work section 2 for further comments to other approaches. While the prior is the major difference with [1] (not to mention our benchmark ‘’CTrL’’), we also propose learning algorithms that are efficient. At test time MNTDP is equivalent to an independent network (this is true also for [1] but not for [2], for instance), and at training time it is a few times bigger than that which easily fits the GPU memory (unlike [1] whose memory grows with the number of tasks at training time, for instance). Recall that the memory we report in table 1 and 2 is the *total* memory used during training by the end of the learning experience; however the memory used during learning each given task is much less in our case (because we always use a very sparse subset of modules). We have added Tab. 10 in the Appendix to show the per-task memory complexity at training and test time. To address the concern about lack of comparison to approaches which evolve the architecture, we have added a comparison to [2] in the revised version of the paper showing that our approach is more efficient on all the streams (see Table 1 and 2, Figure 1, 3 and 4). Since the publicly available implementation of [3] is restricted to MLP architectures applied to MNIST-like datasets, we have performed a comparison on the permuted MNIST dataset (see Table 11 in Appendix), and we also included the RCL as per R3’s suggestion (see Fig. 3). We could not compare to [1] because their code is not publicly released yet. Please, let us know if we have adequately addressed your concerns in terms of contribution and comparison to prior art, and thank you again for your suggestions.
>
> - “Because some of the methods does not add additional parameters for each of the tasks, it is hard to see if the gain from the proposed method or is more of a matter of adding additional parameters to the model. “: That is a very good point. In the paper, we compare our method to other baselines that increase their capacity at each new task (e.g Progressive Neural networks) and also added a comparison to the model [2] with different architectures (HAT and HAT-wide see Section 5.2) corresponding to different capacity levels. In particular, HAT-wide is as big as our final predictor (same final capacity). Our experimental results show that the effectiveness of our approach is not only due to the addition of new parameters but also to the data-driven prior we are using, as we consistently achieve better average accuracy and require less computation (particularly on the long stream), see Fig. 3 and 4, and Tab. 1 and 2. Essentially, methods that do not restrict the search space (like RCL, Xu et al. 2018) cannot scale effectively to S^long, they either run out of memory or become excruciatingly slow as they get older. MNTDP instead uses constant resources over time (always the same number of modules per task both during training and testing). This is why the data driven prior is a very simple and yet very important contribution.

---

> > ### Author Response · Authors · 2020-11-21
> > **Detailed answer to the specific points you raised. (2/2)**
> >
> > - “No positive backward transfer”: That is correct and we clearly state this in the revised version of the paper “Note that only the parameters of the newly added modules are subject to training, de facto preventing forgetting of previous tasks by construction but also preventing positive backward transfer.” (sec. 4.1). While we agree with you that this is a limitation, a) we should not be penalized because we have other contributions (CTrL and data driven prior) and other works also freeze parameters (e.g., RCL), and b) it can be addressed with methods that are orthogonal to the one proposed here (data driven prior). Therefore, we do not see this as a major issue and we leave this extension to future work.
> >
> > - “The proposed approach also requires a task identifier to work with.”: That is correct and we clearly state this “Task descriptors are provided to the learner both during training and test time.” (first paragraph of sec. 3), see also our formulation of predictor, f(x,t|S), which depends on the task descriptor.  Frankly, we do not believe there is a particularly  right nor a preferred continual learning setting as of today. We understand the reviewer’s personal preference, but there are a lot of previous studies (including [1], for instance) that used the same learning setting and assumptions as we do here. There are applications where task descriptors are available and others where they are not. If one wants to deploy a spam classifier for more and more countries, user types, etc such descriptors may be available both at training and test time, for instance. As long as we are clear about our assumptions, the setting should be fine and we should not be penalized for using it.
> >
> > - “The authors argue that the optimal model for permuted MNIST and split CIFAR are independent models because each task is distinct. I would argue that…” What we meant is that each of these tasks has so much training data (relative to the complexity of the learning task) that transfer learning is not quite necessary to achieve good performance on these benchmarks.
> >
> > - “It is not clear what algorithm is used in the experiments for the proposed method. Is the data-driven prior always applied?” That is correct. The table and figures in the main paper only report results using the data-driven prior with k=1; in the paper we mention “using old blocks from the past task that is deemed most similar (k = 1, the default setting)...“ (second paragraph of section 5.2). In the last paragraph of section 5.3 “Ablation” we refer to results using k=all (with more detailed results in the Appendix for this setting).
> >
> > Please, let us know if you have remaining concerns. Thank you!

---

> > > ### Comment · AnonReviewer4 · 2020-11-23
> > > **Reply**
> > >
> > > Thanks for the detailed reply. The paper is improved and the results is stronger with the additional baselines.
> > >
> > > As the authors also acknowledged that the novelty of the method lies in the data driven prior. I would suggest the authors structure the paper so that it is clear the data prior and the dataset is the main contribution. Maybe put the modular network section (i.e. 4.1) into the background section. In that way it is more clear regarding the novelty.

---

> > > > ### Author Response · Authors · 2020-11-23
> > > > **Re**
> > > >
> > > > Thank you for your suggestions. We already made some modifications to better emphasize our contributions in the last revision (see the last paragraph of page 1 and section 4.2). We will make the changes you are suggesting in the next revision.

---

### Official Review · AnonReviewer1 · 2020-10-29
**This paper provides metrics, benchmarks, and algorithms that can evaluate transferability, not just catastrophic forgetting in continual learning. These methods are reasonable, but there seems to be a need for more justification for the five stream cases used in the benchmark for transfer evaluation. This is the main reason for not giving this paper a higher score.**

**Rating:** 7
**Confidence:** 3

**Review:**

This paper provides metrics, benchmarks, and algorithms that can evaluate transferability, not just catastrophic forgetting in continual learning. These methods are reasonable, but there seems to be a need for more justification for the five stream cases used in the benchmark for transfer evaluation. This is the main reason for not giving this paper a higher score.

This paper provides a set of benchmarks for evaluating continual learning algorithms in terms of transfer and scalability, in addition to catastrophic forgetting, which has been mainly used in the past. In particular, five specific task streams were defined, and it is argued that these can be used to assess the transferability, which was not distinguished in the existing benchmarks. Additionally, the authors propose MNTDP, analogous to the modular network, as a new continual learning algorithm to better meet the metrics assessed in the new benchmark.

This paper is well-organized and well-written. Defining metrics for continual learning in Section 3 is not very new, but I think it is necessary. I personally like the authors' claim that transferability should also be an important measure in continual learning. However, there seems to be something that needs to be clearer in terms of motivation in the part that proposes five task streams for benchmarking. It would have been nice if there was a discussion on whether the 5 cases provided as a measure for transfer overlap each other or whether there is an area that cannot be covered. For example, in the case of (MNIST -> CIFAR-10 -> ... -> F-MNIST), it is unclear where it corresponds to the five stream cases. It is also unclear how much the performance improvement of the T(S^{}) values in Table 1 actually represents. I think that referencing the transfer learning benchmarks can help both justify stream case classification and compare transfer performance results.

Another question is, in the case of scalability, what does it mean for regularization-based or replay buffer -based rather than modular network-based continual learning?

---

> ### Author Response · Authors · 2020-11-21
> **We thank you for your review and remarks. We have submitted a new version of the paper (see comment to all reviewers). We provide below here a detailed answer to the specific points you raised. (1/2)**
>
> - “clearer in terms of motivation in the part that proposes five task streams for benchmarking.” Our main motivation is to define new models that leverage past tasks to reduce sample complexity (first paragraph of the introduction). It is therefore important to be able to measure the ability of CL models to transfer and not just to remember. Transfer is usually measured quite naively in the existing literature and we are not aware of any control study on transfer learning, but please let us know if you know of any and we will be happy to include it. One of our objectives and also main contributions is to introduce a new benchmark (new metrics and new streams/datasets) to evaluate CL models on  several transfer dimensions. As it is now better explained (see section 3 below “Streams”), we are not claiming to cover the whole space of possible ways to transfer, but focus on some very basic properties that arguably any good model should possess.
> As shown in the different plots (see for instance the newly added Fig. 1), these different metrics provide a more fine-grained evaluation of the different models by capturing different characteristics. In the future, we would like to extend CTrL to capture other properties of CL models. We have updated section 1 and 3 accordingly.
> - “5 cases provided as a measure for transfer overlap each other or whether there is an area that cannot be covered. “: (see updates in Section 3) We agree that we do not cover the whole space of possible ways to transfer, particularly so because we do not have control over the data generation process (we made the choice to use image datasets to build our streams instead of toy ones where we could control the data generation process at the price of realism) and do not know how and to which extent various tasks relate to each other. Therefore, we consider a handful (5) pretty diverse datasets, and construct streams that check very basic properties that all CL models should possess (but others may most certainly exist), namely: direct transfer (S^-), ability to overwrite knowledge (S^+), ability to transfer to related tasks that only differ in the input (S^in) or output distribution (S^out), and plasticity i.e ability to learn a new task despite having learned before rather unrelated tasks (S^pl). Once again, we expect to include more streams and more metrics in future versions of the benchmark but we believe that the current version already allows a finer-grained evaluation of CL models which is an interesting and useful contribution.
> We have revised the paper in section 3 accordingly. For instance, we say now “While other dimensions certainly exist, here we are focusing on basic properties that any desirable model should possess.” and have better labeled the streams to aid their interpretation.
> - “ It is also unclear how much the performance improvement of the T(S^{}) values in Table 1 actually represents. “ It’s still an open question how to measure transfer. Here we opt for a relative accuracy gain (eq. 3), the difference of accuracy of the same algorithm on the last task when applied to the full stream VS the stream that contains only the last task. In section 3 we say “We would expect this quantity to be positive if there exist previous tasks that are related to the last task. Negative values imply the model has suffered some form of interference or even lack of plasticity when the predictor has too little capacity left to learn the new task.“ Another interpretation is in terms of gain/loss w.r.t. the independent net baseline.

---

> > ### Author Response · Authors · 2020-11-21
> > **Detailed answer to the specific points you raised. (2/2)**
> >
> > - “ referencing the transfer learning benchmarks can help both justify stream case classification and compare transfer performance results.” : Few benchmarks have been proposed in the literature. They usually evaluate fewer dimensions and are usually based on much simpler datasets (MNIST-like images). For instance, https://arxiv.org/pdf/1810.13166.pdf proposes to evaluate CL models over multiple dimensions, but rely on a single and simple stream of tasks based on CIFAR. Another recent example is https://arxiv.org/pdf/1805.09733.pdf which is based on MNIST variations only. We expect the CTrL benchmark to be the first simple-to-use CL benchmark allowing a fine-grained evaluation of multiple transfer dimensions of CL. We are releasing a repository just for CTrL to encourage the community to adopt it irrespective of the modeling choice, and a separate repository for MNTDP.
> >
> > - “ scalability, what does it mean for regularization-based or replay buffer -based rather than modular network-based continual learning?” : Methods that do not grow with the number of tasks are trivially scalable in terms of memory as that is constant with the number of tasks. These methods fall on one side of the spectrum. Independent networks/PNNs are on the other hand, with a linear growth (or more) with the number of tasks. Our objective was thus to propose a simple method that strikes something in between, assuming some tasks relate to each other. Note that other baselines have been added in the last version of the paper.
> >
> > Thank you!

---

### Author Response · Authors · 2020-11-21
**To all reviewers**

Based on the reviews, we have updated the paper as follows:
- We clarified the writing when it was necessary. In particular, we have clarified that our main contribution in terms of modeling is the “data driven prior” which is used to restrict the search space in order to scale to long streams.
- We updated the names of the newly introduced metrics to ease interpretation and increase readability.
- We updated the related work section to include the recommended papers on modular/evolving architectures, and to better describe the differences with our approach.
- We performed additional experiments and compared against the following prior work:  HAT (Serra et al. 2018), DEN (Yoon et al. 2018) and RCL (Xu et al 2018).
- We added further details in the Appendix. In particular, we added a table with the memory complexity both at training and test time, to illustrate the efficiency of our method at inference time.

See below our answer to each reviewer.

---

### Decision · Program_Chairs · 2021-01-07
**Final Decision**

**Decision:**

Accept (Poster)

**Comment:**

This paper presents a new benchmark for evaluating continual learning(CL) algorithms on transferability and scalability. It also introduces a data-driven prior to reduce the architecture searching space. Experiments show the new benchmark helps to analyze the properties of CL algorithms and the proposed algorithm performs better than baselines.

The reviewers raised concerns about evaluation metric, weak baselines, and limited experimental cases for evaluating transferability. The authors added more experiments with stronger baselines and revised the paper based on the reviewers' suggestions. However, the authors also admit that how to evaluate transferability is still an open question.

Despite the concerns, the reviewers generally agreed that the paper is well written,
and the new benchmark is an important contribution for evaluating continual learning algorithms on transferability and scalability.  Hence it makes a worthwhile contribution to ICLR and I'm recommending acceptance of the paper.